# Iron commensalism of mesenchymal glioblastoma promotes ferroptosis susceptibility upon dopamine treatment

Vu T. A. Vo[1,2,3], Sohyun Kim[4], Tuyen N. M. Hua[1,3], Jiwoong Oh[5] & Yangsik Jeong [1,2,3,6,7✉]

The heterogeneity of glioblastoma multiforme (GBM) leads to poor patient prognosis. Here, we aim to investigate the mechanism through which GBM heterogeneity is coordinated to promote tumor progression. We find that proneural (PN)-GBM stem cells (GSCs) secreted dopamine (DA) and transferrin (TF), inducing the proliferation of mesenchymal (MES)-GSCs and enhancing their susceptibility toward ferroptosis. PN-GSC-derived TF stimulates MES-GSC proliferation in an iron-dependent manner. DA acts in an autocrine on PN-GSC growth in a DA receptor D1-dependent manner, while in a paracrine it induces TF receptor 1 expression in MES-GSCs to assist iron uptake and thus enhance ferroptotic vulnerability. Analysis of public datasets reveals worse prognosis of patients with heterogeneous GBM with high iron uptake than those with other GBM subtypes. Collectively, the findings here provide evidence of commensalism symbiosis that causes MES-GSCs to become iron-addicted, which in turn provides a rationale for targeting ferroptosis to treat resistant MES GBM.

---

[1] Department of Biochemistry, Wonju College of Medicine, Yonsei University, Wonju, Gangwon-do, Republic of Korea. [2] Department of Global Medical Science, Wonju College of Medicine, Yonsei University, Wonju, Gangwon-do, Republic of Korea. [3] Mitohormesis Research Center, Wonju College of Medicine, Yonsei University, Wonju, Gangwon-do, Republic of Korea. [4] Department of Physiology, Yonsei University College of Medicine, Yonsei University, Seoul, Republic of Korea. [5] Department of Neurosurgery, Severance Hospital, Yonsei University, Seoul, Republic of Korea. [6] Institute of Lifestyle Medicine, Wonju College of Medicine, Yonsei University, Wonju, Gangwon-do, Republic of Korea. [7] Institute of Mitochondrial Medicine, Wonju College of Medicine, Yonsei University, Wonju, Gangwon-do, Republic of Korea. ✉email: yjeong@yonsei.ac.kr

Glioblastoma multiforme (GBM) is one of the most devastating malignancies due to extremely poor prognosis and obviously major deterioration of patients' life quality associated with motor and cognitive deficits. The condition has an incidence of 3.22 per 100,000 individuals, and even with postoperative, concomitant treatment of radiation and temozolomide, patients with grade IV astrocytoma in general show fifteen months of median survival and approximately 5.5% of a 5-year survival rate[1–4]. Tumor recurrence owing to acquired drug resistance may be attributed to intra-tumoral heterogeneity potentially arising due to phenotypic cellular transitions between GBM subtypes[5–9].

Arising from the subventricular zone, tumor initiating GBM develops into a dominant or mixed tumor of four subtypes, namely classical, neural, proneural (PN), and mesenchymal (MES), all of which exhibit distinct gene expression signatures and thus pathological features[10–13]. The latter two subtypes are more frequent than the others, and MES GBM is generally more aggressive[5–8]. Recent single cell sequencing and anatomical studies have revealed that patients with mixed PN and MES GBM subtypes have the worst survival among all GBM subtypes, with PN GBM being preferentially localized to vascular regions, while MES GBM has extensive hypoxic regions[14,15]. It is conceivable that both subtypes are functionally inter-related or even act symbiotically for GBM progression. For example, extracellular signaling factors derived from apoptotic cells or other cells within the GBM tumor microenvironment (TME) can be transferred and thus promote recipient cell proliferation[16,17]. However, whether this phenomenon occurs between PN and MES GBM cells remains unknown.

Intracellular iron homeostasis is essential for various biological processes including cell proliferation, metabolism, electron transfer, and reactive oxygen species (ROS) production[18]. However, intracellular iron homeostasis should be tightly controlled in cancer cells, otherwise non-bound, free irons biochemically bring about lethal cellular toxicity to trigger iron-dependent non-apoptotic cell death, which is also known as ferroptosis[19]. Thus, the targeting ferroptosis may hold therapeutic potential for various relapsed, mesenchymal malignancies including GBM[20,21].

The neurotransmitter dopamine (DA) is a major catecholamine synthesized in the substantia nigra of the midbrain where dopaminergic neurons express the rate-limiting enzyme, tyrosine hydroxylase (TH), which converts tyrosine to dihydroxyphenylalanine (DOPA)[22]. Interestingly, recent studies have shown that GBM cells synthesize DA, which is implicated in the tumorigenesis and phenotypic transition of GBM[23,24]. Furthermore, DA promotes the uptake of non-transferrin (TF)-bound iron into macrophage and subsequently increases intracellular oxidative stress in a DA receptor D5 (DRD5)- or monoamine oxidase-dependent manner[25]. In addition to inducing various alterations in molecular pathways required to cope with unfavorable stress conditions, GBM stem-like cells preferentially uptake iron through a canonical TF–TF receptor 1 (TfR1) pathway, thereby promoting tumorigenesis[26].

In the present study, we sought to explore the mechanism through which GBM heterogeneity is coordinated to enable tumor progression. We found that PN glioblastoma stem cells (GSCs) secreted both DA and TF, which in turn supported the proliferation of neighboring MES GSCs. DA enhanced iron uptake in MES GSCs, which subsequently became addicted to iron for proliferation, and paradoxically, vulnerable to ferroptosis. The findings of the present study provide biological evidence of intra-tumoral commensalism in heterogeneous GBM, and further an insightful rationale of targeting ferroptosis for treating GBM.

## Results

**PN GSCs secrete TF and DA.** To explore factors involved in the symbiotic co-existence of PN and MES tumors, we first examined the growth response of MES GSCs in PN GSCs-conditioned media or vice versa (Supplementary Fig. 1a). All MES GSCs exhibited a significant beneficial growth response in PN GSC-conditioned media, while no growth response was observed in PN GSCs under MES GSC-conditioned media (Fig. 1a, upper and middle). Representative images of the growth response of PN 448T and MES 83 cells under conditioned media are provided (Fig. 1a, lower). This was further confirmed by direct co-culture of PN and MES cells using permeable insert (Fig. 1b, upper) or being labelled with different fluorescence probes and let them grow in the same culture plate (Fig. 1b, lower). While PN X02 remained unaffected, MES 83 cells consistently showed more favorable growth in the presence of PN X02. These observations suggested that PN GSCs secreted humoral factors, potentially supporting the growth of MES GSCs. We sought to identify the specific growth-promoting factors present in PN GSC-conditioned media. We analyzed a previously published RNA sequencing dataset[9] and an in silico database, narrowing our search to two candidates, TF and DA, as potential factors conferring the above-described unidirectional growth advantage. We then determined the expression of TF, TfR1, and DA-related proteins in the GSC panel. PN GSCs showed high expression and secretion of TF, while substantially higher TfR1 expression was observed in MES GSCs than in PN GSCs (Fig. 1c, Supplementary Fig. 1b), which was also noted in the previously mentioned RNA sequencing dataset (Fig. 1d, upper). To objectively confirm the in vitro results, we carried out in silico analysis of the dataset of GBMs from Ivy Glioblastoma Atlas Project (Ivy GAP, http://glioblastoma.alleninstitute.org/). Note that the Ivy GAP is an RNA-seq database from GBM tissues, which are anatomically microdissected into multiple regions: the leading edge (LE) and infiltrating tumor (IT) which represent the PN subtype in a normoxic TME, and the cellular tumor (CT), pseudopalisading cells around necrosis (CTpan), and microvascular proliferation (CTmvp), which are representative of the MES subtype (Supplementary Fig. 1c). Expression of HIF1α target genes identified the hypoxic region potentially associated with MES GBM (Supplementary Fig. 1c). To our surprise, the PN LE region exhibited the highest expression of TF, which gradually decreased toward central regions, with CTmvp region showing the lowest expression, while TFRC expression followed an opposite pattern, except in the CTmvp region (Fig. 1d, middle and lower). Other genes associated with iron trafficking and storage were also exclusively expressed in the CTpan region of MES GBM, while the iron exporter, SLC40A1, was expressed within the microvascular CTmvp region (Supplementary Fig. 1d, upper). Similarly, by assessing the expression of critical enzymes for DA metabolism, we found that PN GSCs autonomously biosynthesize and secrete DA, which may support GBM heterogeneity (Fig. 1e; Supplementary Fig. 1d, lower). Gene set enrichment analysis (GSEA) using the same dataset confirmed the upregulation of genes associated with dopamine_secretion and transport pathways in the PN regions (Supplementary Fig. 1e). Phenylalanine hydroxylase (PAH), TH, and DOPA decarboxylase (DDC) are enzymes involved in sequential biochemical reactions converting phenylalanine or tyrosine to L-2,4-dihydroxyphenylalanine (L-DOPA) and, finally, DA. DRD5 expression was high in PN GSCs but not in MES GSCs (Fig. 1e, Supplementary Fig. 1f), suggesting that the secreted DA may act on PN subtype cells in an autocrine manner through this receptor. We next determined the factors that may regulate TF and DA secretion from PN. Given that iron concentration would be one of the environmental factors regulating TF secretion[27], we found treatment of low iron concentration

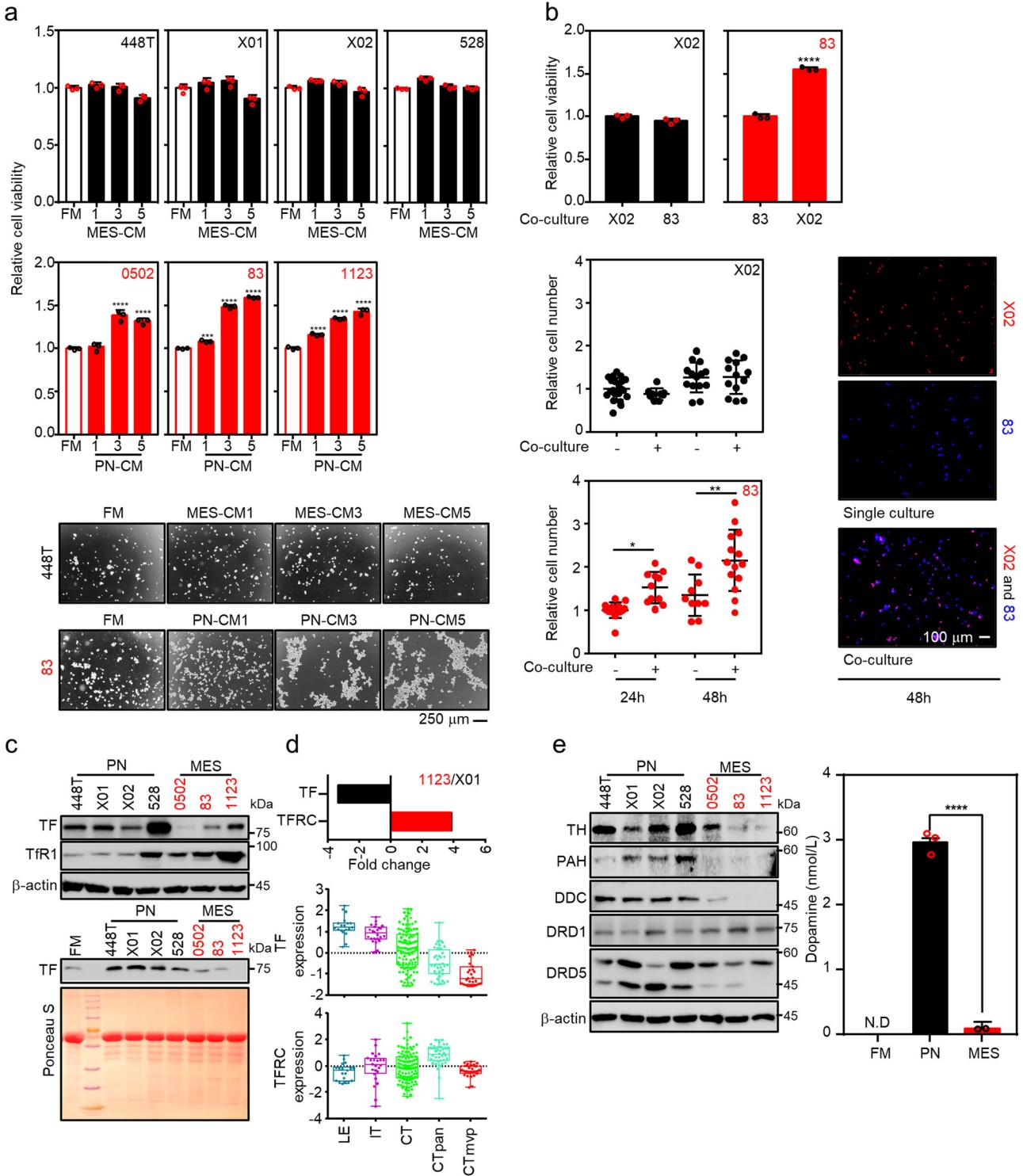

increased TF secretion in the media, while high iron concentration reversed this (Supplementary Fig. 1g). In addition, considering that calcium could be one of the crucial environmental factors stimulating DA secretion[28–32] in cancer cells[28,29], we found DA secretion from the PN-GSCs is dependent on extracellular calcium by carrying out calcium depletion experiment (Supplementary Fig. 1h). Using GBM patient tissues, we next sought to confirm the results of in vitro expression of proteins involved in iron uptake as well as DA biosynthesis (Fig. 2). GBM tissue samples (GBM1211, GBM2021) were prospectively acquired and anatomically microdissected into center (C) and

margin (M) or peripheral regions followed by immunostaining for TF, TfR1, DRD5, TH, PAH, and DDC. The two microdissected regions were molecularly characterized for the expression of HIF1α target genes and SOX2 or CD44, as PN and MES markers, respectively (Fig. 2a). Consistent with the in vitro results, TF, DRD5, TH, PAH, and DDC were highly expressed in the peripheral PN region, while TfR1 expression was higher in the central MES region (Fig. 2b). Unlike DRD5, other DRDs receptor were not expressed in GBM tissues (Supplementary Fig. 2a). In addition to the anatomical expression profile, a set of paraffin-embedded GBM patient tissues were retrospectively obtained

**Fig. 1 PN GSCs secrete DA and TF. a** PN GSC-conditioned culture medium induces MES GSC proliferation but not vice versa. (upper and middle) PN or MES GSCs were cultured in the indicated conditioned media for 3 days, followed by MTS assay for cell viability analysis. (lower) Representative images of MES 83 and PN 448T GSCs showing the growth response in a particular conditioned medium. Three different conditioned media were prepared at 1, 3, and 5 days after culturing the corresponding GSCs. **b** PN GSCs supports MES GSC proliferation in co-culture models. (upper) PN and MES GSCs were separately co-cultured in top and bottom compartments or vice versa using permeable insert for 3 days, followed by MTS assay. (lower) PN X02 and MES 83 GSCs were labeled with CMTPX red and Hoechst 33342, respectively, followed by direct co-culturing for 24 h or 48 h. **c, d** TF is secreted from PN GSCs. **c** Protein expression of TF and TfR1 and TF secretion in the GSC panel. (upper) Basal level of TF and TfR1 protein expression is shown in the GSC panel. (lower) Conditioned medium from PN GSCs cultured for 3 days was assayed for the secretion of TF via western blotting. **d** Expression of *TF* and *TFRC* is presented from RNA sequencing results in MES 1123 vs. PN X01 cells or from the Ivy GAP, a publicly available RNA-seq database of GBM tissues anatomically microdissected into leading edge (LE), infiltrating tumor (IT), cellular tumor (CT), pseudopalisading cells around necrosis (CTpan), and microvascular proliferation (CTmvp). **e** PN synthesizes and secretes DA. Expression of DA synthesis related enzymes (left) and DA secretion (right) were analyzed using western blotting and ELISA for determining the levels in conditioned media. (CM, conditioned media; FM, fresh media). Error bars represent standard deviation from the mean.

from a tissue bank and further assayed for intra-tumoral differences in the expression of genes of interest, confirming the association of PN regions with DRD5 and TF as well as TfR1 expression in MES regions (Fig. 2c, d). Further in silico analysis using multiple GBM datasets obtained from Gliovis project (http://gliovis.bioinfo.cnio.es) confirmed that PN cells increased expression of TF gene, DA receptors and synthesis enzymes whereas MES upregulated transferrin receptor TFRC (Supplementary Fig. 2b, c). Taken together, these data support the notion of intra-tumorigenic growth benefit for GBM progression in which particularly provide an example of commensal symbiosis of MES dependent on the two humoral factors from the neighboring PN GSCs.

**MES GSCs show preferential iron uptake leading to cell proliferation**. Having identified the secretion of TF and DA from PN GSCs, we next explored the biological mechanisms involved in supporting MES GSC proliferation. First, all GSCs were assessed for their dependence on iron for cellular growth through the addition of exogenous TF or ferric ions in the culture media. Both TF and ferric ions considerably enhanced the growth of MES GSCs but not that of PN GSCs. This effect was markedly more pronounced, especially under hypoxic conditions than under normoxic conditions, from which MES GSCs are considered to originate[15,33,34] (Fig. 3a). The iron dependency of MES GSC growth was further confirmed, as MES GSCs were more susceptible to the TfR1 pharmacological inhibitor, ferristatin II, than PN GSCs (Supplementary Fig. 3a). In addition, treatment with an anti-TF neutralizing antibody reversed TF-induced MES GSC growth, suggesting that PN GSC-derived TF contributes to MES GSC proliferation (Fig. 3b). Mechanistically, dose-dependent TF treatment enhanced the phosphorylation of the oncogene, Src, and its downstream component, ERK, which are potentially responsible for MES GSC proliferation but not PN GSC proliferation (Fig. 3c, Supplementary Fig. 3b). While it has been previously reported that colorectal tumor cells take up iron via divalent metal transporter 1, subsequently activating Janus kinases-signal transducer and activator of transcription proteins 3 (STAT3) signaling, MES GSCs exhibited no STAT3 signaling activation even with upregulated Src phosphorylation (Supplementary Fig. 3b)[35]. Likewise, extracellular iron treatment induced TfR1 expression in MES GSCs in a dose-dependent manner, while PN GSCs exhibited no response (Fig. 3d). This resulted in the intracellular accumulation of iron in MES GSCs following iron treatment, while no or low iron uptake in PN GSCs was observed (Fig. 3e). Such accumulation activated oncogenic Src signaling in MES GSCs but not PN GSCs. It has been previously reported that pSrc expression increases upon TF treatment, and the activated Src is involved in iron uptake[36]. However, in the present study, it was the uptake of iron, rather than TF, which

induced Src activation as PN GSCs showed no dose-dependent pSrc activation following TF treatment. Indeed, MES, but not PN, GSCs exhibited increased iron uptake under the addition of exogenous iron in the culture media (Fig. 3e). As MES GSCs are located in the hypoxic region of patient tumor tissues[15], we examined the growth response of the GBM cell panel to the same treatment conditions under hypoxia. MES GSCs showed a more pronounced growth response to TF and ferric ions under hypoxia than normoxia (Fig. 3a). Interestingly, both treatments increased the expression of HIF1α and its target genes only in MES GSCs and not in PN GSCs (Fig. 3f, Supplementary Fig. 3c). Given that iron stimulation induced Src-ERK signaling (Fig. 3c, Supplementary Fig. 3b) and that may be important for maintaining hypoxia-inducible factor Hif1α[37–39], two different Src inhibitors, SU6656 and PP2, were tested for the downstream signaling. We found PP2 inhibition of downstream ERK, but not SU6656 inhibition of downstream STAT3, was able to abolish iron or TF-stabilized Hif1α (Supplementary Fig. 3d). Consistently, PP2 reversed the MES cell growth upon iron or TF treatment (Fig. 3g). Taken together, these results suggest that PN-derived TF may be used preferentially by MES GSCs, thus exemplifying iron commensalism within mixed PN and MES GBM tumors.

**DA acts in an autocrine or a paracrine manner on GBMs**. As PN GSCs-derived TF influenced MES GSCs proliferation in a paracrine manner to enhance iron uptake, we next sought to explore the biological function of DA in sustaining the co-existence of PN and MES subtypes within GBM tumors. As DA function is controversial in the GBM growth depending on expression of DA receptor types[23,24,40–42] and the upregulation of DRD5 expression was observed in PN GSCs, but not in MES GSCs (Fig. 1e), we assessed whether GSCs of the two subtypes would respond differently to treatment with DA or an antagonist of downstream dopaminergic receptor signaling pathways. Indeed, DA treatment considerably induced Src phosphorylation in PN GSCs, with little or no response in the MES GSCs (Fig. 4a). Similarly, PN GSCs, compared with MES GSCs, exhibited increased susceptibility to growth inhibition following treatment with the pan-DA receptor blocker flupenthixol, the DRD1/5-specific inhibitor LE300, or carbidopa, an antagonist for DDC which was highly expressed in PN GSCs (Figs. 1e, 4b). Note that MES 0502 cells exhibited high TH and moderate DDC expression as well as similar growth inhibition to that observed in PN GSCs upon carbidopa treatment, which supported the notion that GSCs growth inhibition reflected DDC protein expression (Figs. 1e, 4b). Moreover, LE300 treatment suppressed pSrc expression exclusively in PN GSCs (Supplementary Fig. 4a), suggesting the importance of this oncogene for GSC growth and its differential DA- or iron-induced activation between PN and MES GSCs, respectively. Along with the pharmacological approach, DRD5

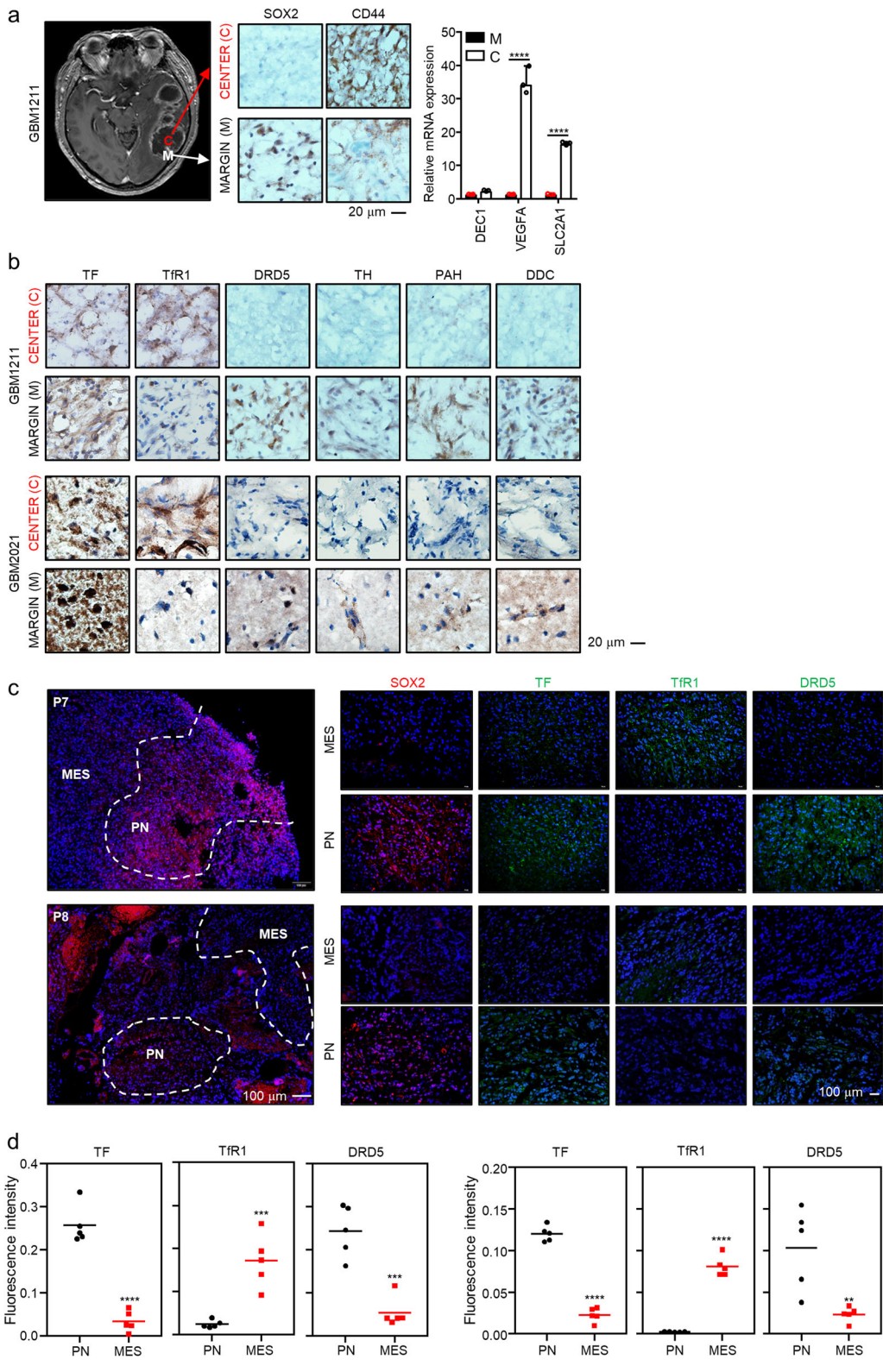

**Fig. 2 Iron and DA-related protein expression in GBM patient tissues. a, b** IHC for proteins and mRNA expression of interest in the margin (M) and center (C) regions of the brain tissues. **a** A MRI image represents patient GBM1211 (left). Represented are IHCs (middle) for SOX2 and CD44 proteins, and mRNA expression (right) for *DEC1, VEGFA,* and *SLC2A1* in margin (M) and center (C), respectively. **b** Immunohistochemistry of TF, TfR1, DRD5, TH, PAH, and DDC. **c** Immunofluorescence results for genes of interest in representative human brain tissues. **d** Quantification of fluorescence signal from **c**. Error bars represent standard deviation from the mean.

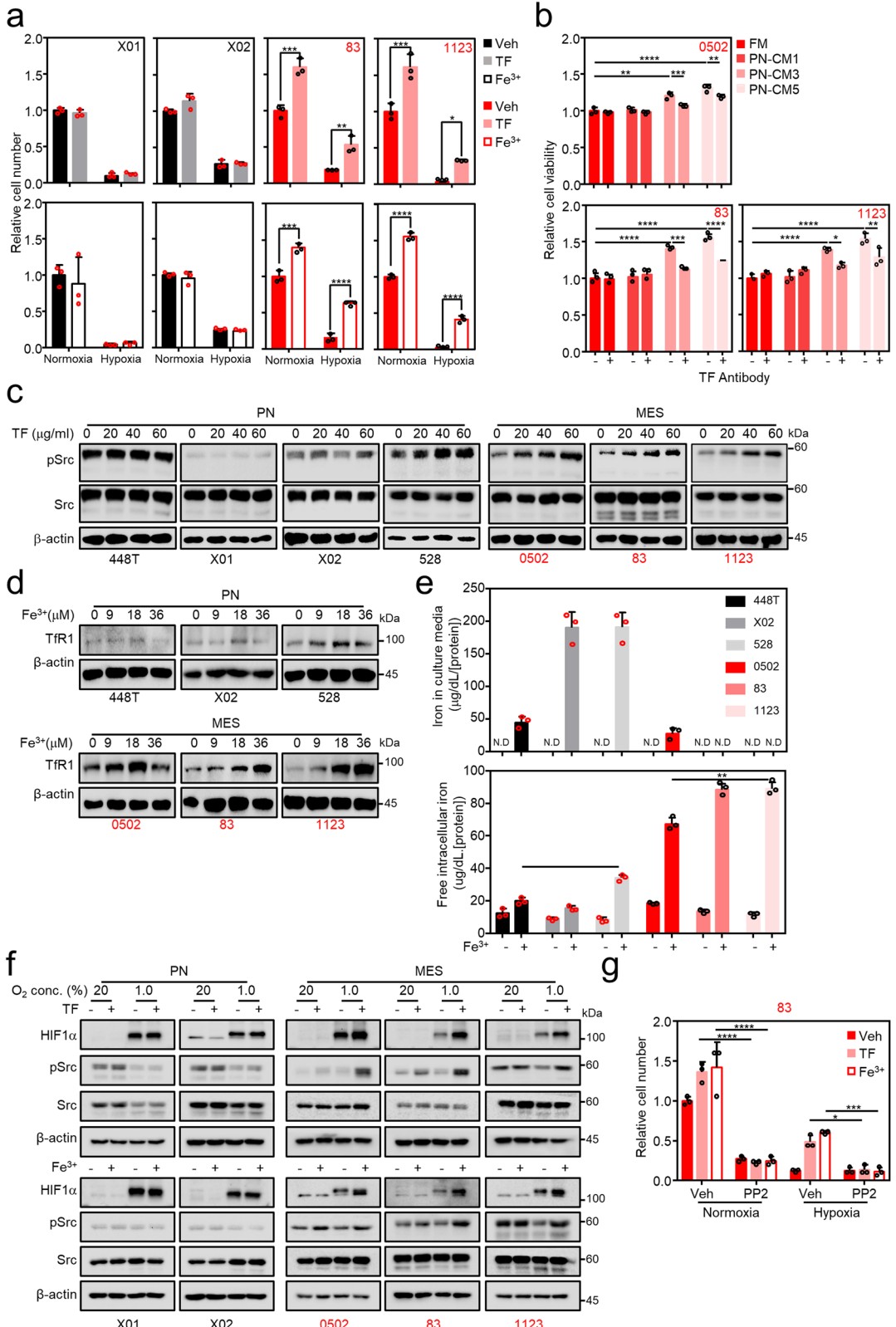

knowdown showed growth-inhibition of PN but not MES GSCs by decreasing Src activation upon DA treatment, (Supplementary Fig. 4b, c) suggesting the autocrine function of DA is mediated by DRD5 in PN GSCs. Together with the pharmacological results, we further examined the growth dependency of PN GSCs in phenylalanine- and/or tyrosine-deprived culture medium. PN GSCs were auxotrophic for the precursor amino acids necessary

for DA biosynthesis, while MES GSCs were not (Fig. 4c). Furthermore, the auxotrophy of PN GSCs for Phe and Tyr was rescued by adding DA to the conditioned media (Supplementary Fig. 4d). These results suggest that PN GSCs endogenously synthesize and secrete DA, contributing to the maintenance of proliferation in an autocrine manner. We further considered whether DA paracrine signaling was involved in the uptake of iron by

**Fig. 3 TF induces MES GSC proliferation via preferential iron uptake. a** MES GSC proliferation, but not PN GSC proliferation, increased upon TF or ferric ion ($Fe^{3+}$) treatment. In vitro proliferation of the GSC panel was measured using MTS assay at 3 days after incubation in TF (20 μg/ml) or 18 μM of $Fe^{3+}$ in conditioned culture media. Cell viability was assayed under normoxic or hypoxic (1% $O_2$) conditions. **b** Reversed MES GSC growth via treatment with anti-TF neutralizing antibodies. In vitro proliferation of MES GSCs was assessed via MTS assay at 3 days after culturing in PN GSC-conditioned media with or without 0.25 μg/ml of anti-TF neutralizing antibodies. **c** Oncogenic Src phosphorylation after TF treatment in GSCs. The GSC panel was assayed for Src phosphorylation following treatment with different concentrations of TF. **d** TfR1 expression after ferric ion treatment. TfR1 expression was identified in the GSC panel after treatment with different doses of ferric ($Fe^{3+}$) ions. **e** Increased total iron uptake in MES GSCs. Iron uptake was measured at 24 h after GSCs were incubated in conditioned media with ferric ($Fe^{3+}$) ions. **f** TF and $Fe^{3+}$ stabilize HIF1α and pSrc in MES GSCs, but not PN GSCs, under hypoxia. Under normoxic and hypoxic conditions, GSCs were treated with 20 μg/ml TF (upper) or 18 μM of $Fe^{3+}$ (lower) for 3 days, followed by immunoblotting for HIF1α, Src, and pSrc. **g** MES 83 growth response upon treatment of TF, $Fe^{3+}$, and Src inhibitor PP2. MES 83 was treated with 20 μg/ml TF or 18 μM of $Fe^{3+}$ with or without 20 μM of PP2 under normoxic and hypoxic condition. Cell growth were quantified by direct cell counting. Error bars represent standard deviation from the mean.

MES GSCs. DA treatment increased TfR1 expression (Fig. 4d, Supplementary Fig. 4e), and it enhanced total iron uptake in MES GSC, but not PN GSCs, in a dose-dependent manner (Fig. 4e) when GSCs were exogenously supplemented with $Fe^{3+}$. Further, MES GSCs exhibited a significant increase in both total iron uptake and intracellular labile iron ($Fe^{2+}$) pool, whereas PN GSCs showed an increased intracellular labile iron pool but exhibited no change in total iron uptake following DA treatment (Fig. 4e). This might be due to the unaffected or even decreased TfR1 availability following autocrine DA signaling in PN GSCs in contrast to TfR1 induction by paracrine DA in MES GSCs (Fig. 4d).

These results imply that PN GSCs synthesize and secrete DA, promoting the growth of PN GSCs in an autocrine manner. Furthermore, DA and TF synergistically increase iron uptake by MES GSCs in a paracrine manner.

**Dopamine enhances ferroptosis of GBMs.** After observing increased iron uptake into MES GSCs mediated via the paracrine effects of PN GSC-secreted DA and TF, we sought to investigate the biological consequence of intracellular iron accumulation and further link to potential therapeutic strategies for MES GBM. To this end, the GSC panel was pharmacologically assessed for ferroptosis induction. Interestingly, PN GSCs, but not MES GSCs, became susceptible to various ferroptotic stresses, such as RSL3-mediated GPX4 inhibition, erastin-mediated system Xc inhibition, and culture medium with cystine deprivation, and the effects of which were reversed via ferrostatin-1-induced inhibition of lipid peroxidation (Fig. 5a, Supplementary Fig. 5a). Interestingly, MES GSCs became susceptible to ferroptotic stress upon DA treatment (Fig. 5a, Supplementary Fig. 5b, c), which might be associated with increased iron uptake in MES GSCs and the accumulated labile iron pool in both subtypes, as described earlier (Fig. 4e). Notably, all GSCs exhibited increased intracellular ROS generation upon DA treatment, which was rescued by flupenthixol, suggestive of the dependence of ROS generation on dopaminergic receptor function (Fig. 5b). Hydroxyl radicals from the Fenton reaction of the labile iron ($Fe^{2+}$) and intracellular ROS by DA subsequently drive lipid peroxidation (Fig. 5c; Supplementary Fig. 5d, e) as well as increased intracellular 4-hydroxynonenal (4-HNE) level (Supplementary Fig. 5f). To confirm our in vitro results, we established in vivo orthotopic and heterotopic GBM models to evaluate ferroptosis induction as a therapeutic strategy for GBM. The orthotopic mouse model showed longer overall survival in the ferroptosis inducer sorafenib-treated group than the control group. Combined treatment with sorafenib and DA caused a further increase in overall survival compared to treatment with sorafenib alone, while treatment with DA alone did not influence overall survival (Fig. 5d, upper). Note that sorafenib was chosen as a ferroptosis inducer by inhibiting cystine uptake since it is known to cross the

blood-brain barrier and has been clinically approved for treating several therapy-resistant cancers[43–45] (Supplementary Fig. 6). Biochemical analysis of tumor tissues from the orthotopic model revealed that lipid peroxidation increased in the combined treatment group, as indicated by 4-HNE staining (Fig. 5d, lower). Consistently, the heterotopic model showed a significantly decreased overall tumor size and weight upon combined treatment with DA and the ferroptosis inducer erastin (Supplementary Fig. 7a). Furthermore, while erastin alone could induce a modest level of 4-HNE, iron accumulation and strong 4-HNE staining were observed in tumor tissues from the combined treatment group while the MES marker CD44 substantially decreased (Supplementary Fig. 7b, c, upper). Note that all the treatment did not affect mouse body weight (Supplementary Fig. 7c, lower). Based on our current pre-clinical results and those from the previous study on the poor prognosis of heterogeneous GBM, we next proceeded to assess the prognostic value of subtype symbiosis in tissue samples obtained from GBM patients. Considering that TF and DRD5 expression could be a feature of PN cells whereas TFRC may represent MES, we questioned if a mixed-subtype with upregulation of both TF and TFRC or DRD5 and TFRC may have relevant prognosis value, as implied previously[14]. By utilizing datasets publicly available, we performed survival analysis of GBM patients based on the expression of genes of interest. In agreement with the findings of the previous study[26], high TFRC expression was associated with low overall survival, while TF and DRD5 expression had no prognostic relevance (Supplementary Fig. 8). When combined, the groups with high expression of both genes (TF HIGH - TFRC HIGH or DRD5 HIGH – TFRC HIGH) had the worst overall survival among groups (Fig. 5e), indicating function of these genes in heterogenous growth of GBM is potentially associated with poor patients survival. In a panel of GBM patient tissues, TfR1 and DRD5 expression showed a negative and positive correlation with the expression of the PN marker, SOX2, respectively (Fig. 5f), indicating that the differential expression of these two receptors may contribute to the progression of heterogeneous GBM. The hypothetical model of symbiosis between PN and MES GSCs in the maintenance of GBM heterogeneity is illustrated in Fig. 6.

## Discussion
GBM is an extremely aggressive tumor with poor prognosis, attributable to "phenotypic plasticity" accompanying intra-tumoral heterogeneity, which is clinically associated with disease progression and acquired therapy resistance[12,14,15]. Additionally, accumulating genetic mutations in driver oncogenes give rise to diverse tumor clones with unique molecular and genetic signatures, which potentially develop into a single dominant type or mixed subtypes of GBM. In turn, these heterogeneous tumors overcome autonomous or non-autonomous restricted growth

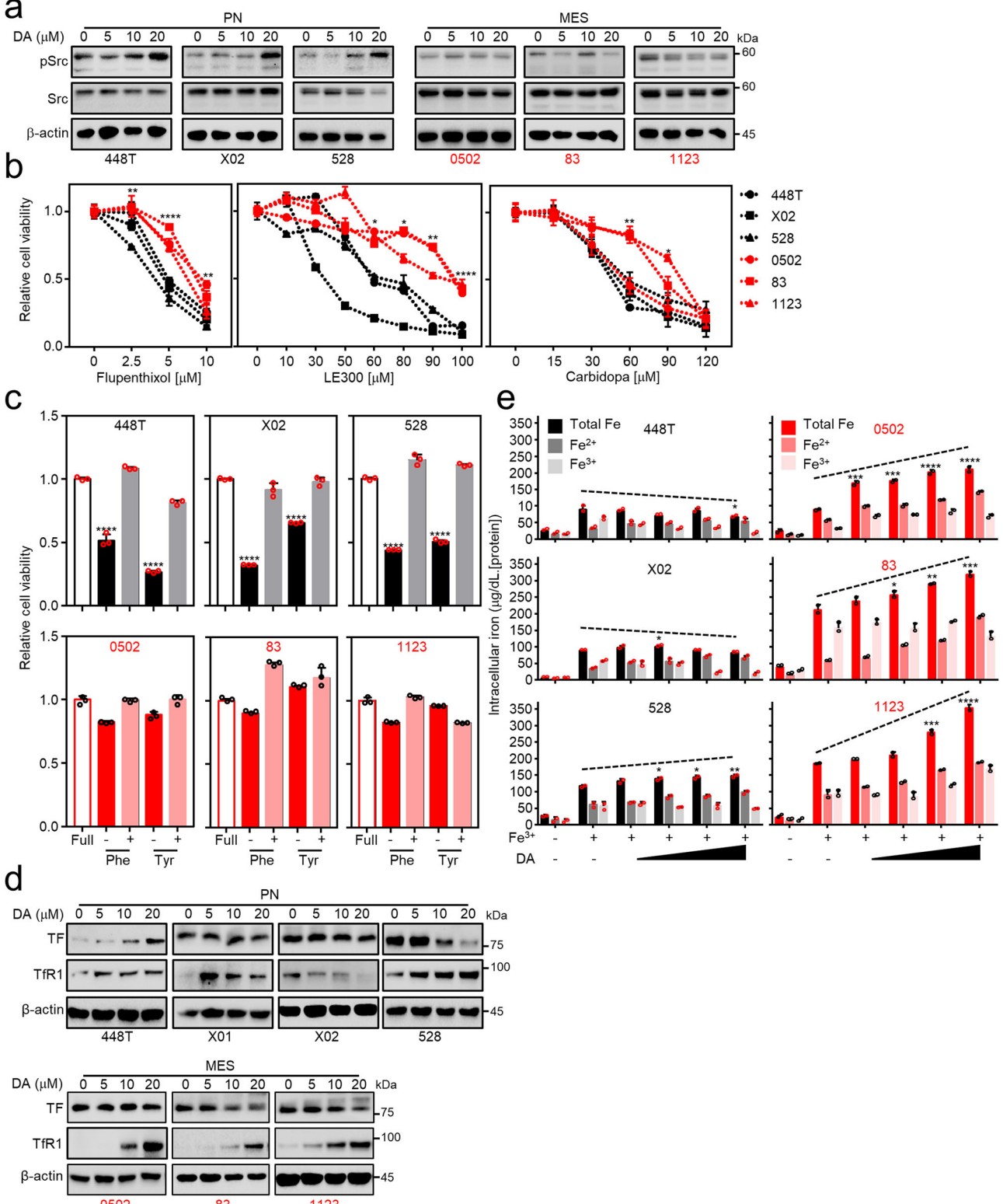

**Fig. 4 DA affects GSCs in an autocrine and paracrine. a** Src phosphorylation by DA treatment in GSCs. The GSC panel was treated with DA at different concentrations, followed by immunoblotting for the proteins of interest. **b** In vitro cell viability of the GSC panel upon treatment with dopaminergic signaling inhibitors. A panel of GSCs was treated with a pan-dopaminergic receptor inhibitor flupenthixol for 2 days, the specific DRD1/5 inhibitor LE300 for 3 days, or carbidopa targeting DOPA decarboxylase for 2 days. Cell viability was measured using MTS assays after treatment of the corresponding drugs. **c** PN GSCs, but not MES GSCs, exhibited auxotrophy for phenylalanine and tyrosine. The GSC panel was cultured in phenylalanine- or tyrosine-deficient medium and then subjected to cell viability assays. **d** Both TF and TfR1 expressions in the GSC panel following treatment with different doses of DA. **e** Measurement of total iron uptake, intracellular ferrous ($Fe^{2+}$) ions, and ferric ($Fe^{3+}$) ions following DA treatment. The GSC panel was incubated in $Fe^{3+}$ conditioned media for 24 h with or without treatment of DA, followed by the measurement of total iron uptake as well as intracellular $Fe^{2+}$ and $Fe^{3+}$ ions. Error bars represent standard deviation from the mean.

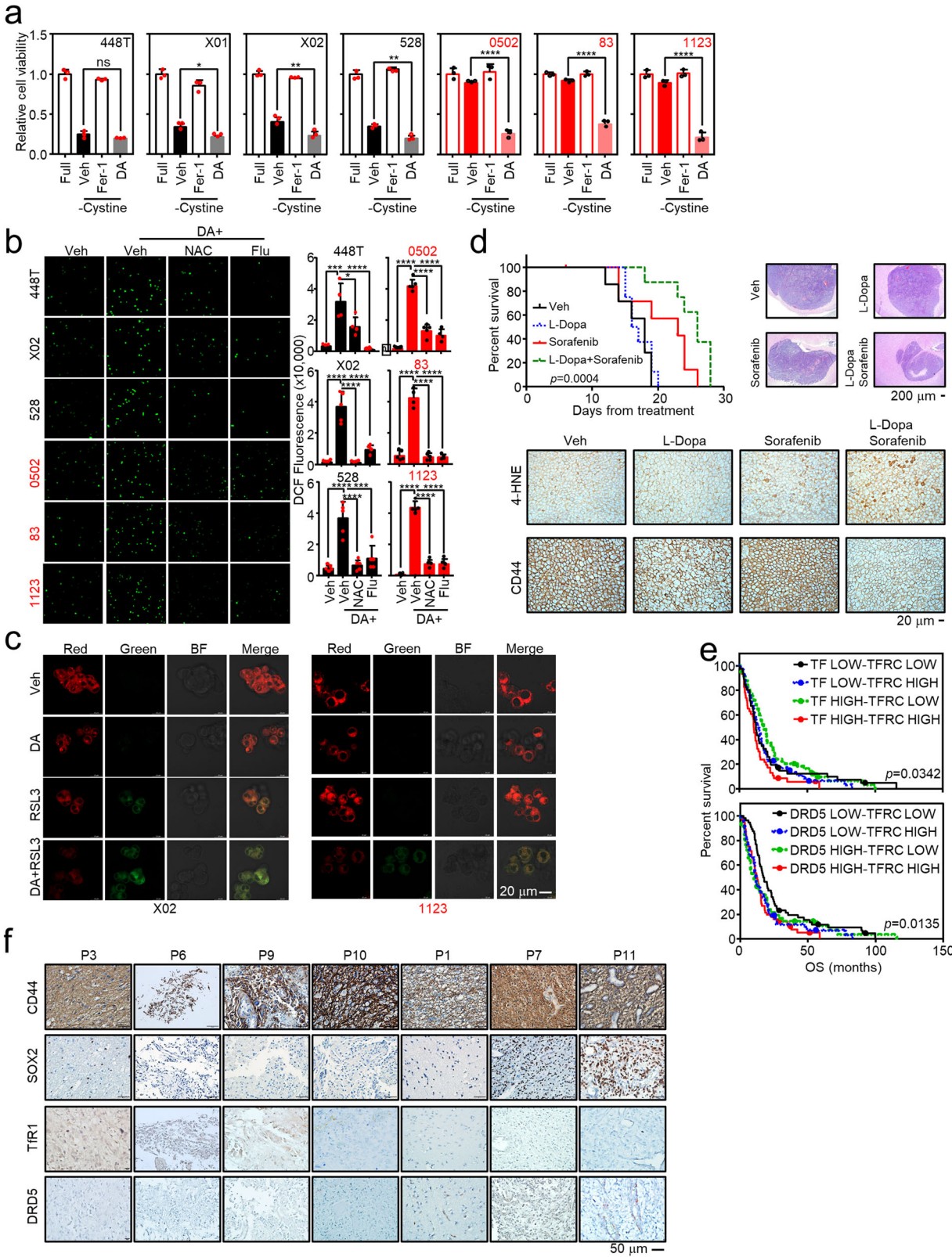

pressure and spread throughout the brain[14,46]. Strong studies have revealed that metabolic reprogramming of individual tumor cells progresses into intra-tumoral metabolic symbiosis described as "mutualism" and "commensalism"[47,48]. This biochemical co-existence contributes to tumor heterogeneity through the exchange of metabolites and signaling factors between tumor cell subtypes[49–52]. Thus, intra-tumoral symbiosis may represent a

suitable target, the perturbation of which can lead to tumor cell death.

Here, we systematically identified molecular factors critical for the maintenance of tumor heterogeneity between PN and MES GBM. To this end, we carried out multiple independent, molecular and biochemical approaches to utilize in vitro, in silico, and in vivo models. We then sought to confirm our observations in

**Fig. 5 DA enhances ferroptosis in GSCs. a** GSCs are susceptible to ferroptotic stress. The viability of GSCs was assessed following cystine deprivation in combination with ferrostatin-1 (Fer-1) or DA using MTS assays at 2 days after treatment with the corresponding drugs. **b** DA treatment induces intracellular ROS generation in GSCs. GSCs were treated with 20 μM of DA with or without N-acetylcysteine (NAC) or flupenthixol (Flu). Fluorescence image of intracellular ROS is shown (left), and DCF-fluorescence is quantified (right). **c** Lipid peroxidation in GSCs upon ferroptosis induction. PN X02 and MES 1123 GSCs were treated with 1 μM of RSL3 in the presence or absence of 20 μM DA, followed by visualization of lipid peroxidation as described in materials and methods. **d** In vivo orthotopic xenograft mouse models. Orthotopic model was established by intracranial injection of MES 83 cells into 5-weeks-old female athymic nude mice. Mice with tumors were randomly divided into four groups and treated with vehicle or indicated drugs. Kaplan–Meier survival curve for mice was shown (left). Orthotopic xenografted tumor tissues are used for H&E staining (upper right), lipid peroxidation 4-HNE immunostaining (middle), and CD44 expression (lower). **e** Iron uptake and DA signaling-related genes are associated with poor patient prognosis. Kaplan–Meier plots for GBM patient survival based on combined *TF* and *TFRC* expression (upper) or *DRD5* and *TFRC* expression (lower). Public TCGA datasets (Nature 2008) were utilized to analyze patient survival for the indicated gene sets. **f** IHCs for proteins of interest in GBM patient tissues. Error bars represent standard deviation from the mean.

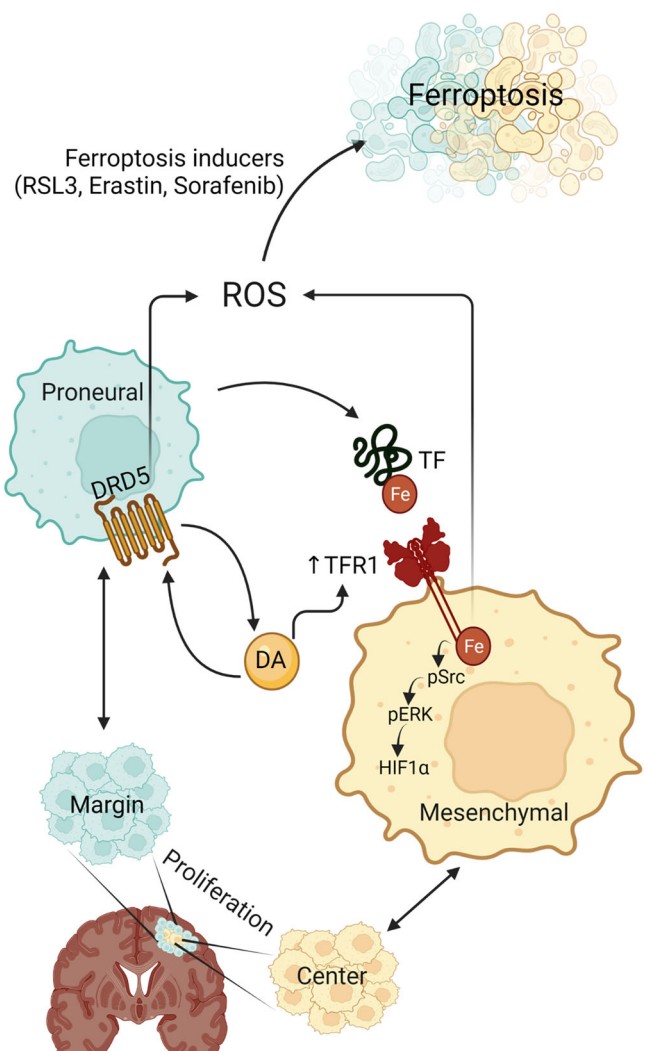

**Fig. 6 Hypothetical model of commensalism symbiosis in heterogeneous GBM.** The figure was created using a program provided by BioRender.com.

human GBM tissues and correlate features to patient survival through the analysis of publicly available datasets. In summary, we found (1) a commensal relationship between PN and MES GSCs, wherein the latter gain tumorigenic benefit from PN GSCs; (2) identified that TF and DA, as humoral factors secreted from PN GSCs, enhance iron uptake by MES GSCs; (3) demonstrated that both GBM subtypes are susceptible to ferroptosis, with DA inducing ferroptotic vulnerability in the iron-addicted MES GBM. Although the present study proposed an interesting symbiotic

mechanism that can be exploited for treating GBM, it would be important to have further discussion on the biological and clinical aspects for a couple of issues raised from this study. First, while we proposed commensalism, mutualism between the two subtypes of GBM cannot be excluded, considering the possibility of direct interactions, rather than humoral signaling, between the tumor subtypes. Many recent studies on other tumor types have reported intercellular interactions supporting cross-benefit between sub-populations in favor of tumor growth[53]. An example includes the secretion of cytokines, growth factors, or angiogenic factors for interclonal cooperation[54]. A decade ago, Sonveaux et al. proposed a model of metabolic symbiosis in which oxidative cells utilize lactate secreted from glycolytic cells under the hypoxic intra-tumoral environment[55]. This would be a case of energetic commensalism between hypoxic and normoxic tumor cells.

Second, iron uptake is crucial for the maintenance of intracellular homeostasis required for tumor cell proliferation, particularly under an ischemic or a hypoxic TME. Several studies have previously reported that high TfR1 expression is associated with poor prognosis in GBM patients[26,56]. In addition, considering the localization of MES GSCs to the hypoxic tumor tissue, we observed that MES GSCs were inherently iron-addicted to properly cope with hypoxic stress. Accordingly, MES GSCs exhibited an increased accumulation of intracellular iron through the collaborative action of PN GSC-derived TF and DA. With respect to the role of DA in tumorigenesis, it is known to be context-dependent based on the expression of receptor subtypes in various cancers including GBM[57,58]. In the present study, expression analysis revealed higher D1-like receptor expression in PN GSCs, whereas D2-like receptors exhibited no difference between PN and MES GSCs. While its effects may be mediated through different DRD subtypes, DA contributes to GBM progression by promoting PN GSC proliferation as well increasing iron uptake that leads to MES GSC growth. Thus, the functional co-operation of DA and TF contributes to overall GBM progression.

Lastly, increased iron uptake may generate cellular vulnerability to ferroptosis, which has gained increasing attention as a therapeutic target in many drug-resistant, MES-type cancers[20]. Herein we linked the iron commensalism of MES GSCs to the iron-dependent, biochemical cell death following extensive lipid peroxidation. PN GSCs were susceptible to ferroptosis induction, whereas MES GSCs exhibited no response. However, the iron addiction of MES GSCs facilitated ferroptosis induction following DA co-treatment. DA induced intracellular ROS generation without pronounced lipid peroxidation. Importantly, its effects on MES GSCs were significantly enhanced upon GPX4 inhibition, implying that DA primed this subtype to become more susceptible to such biochemical stress. Considering the current observations and findings of previous studies that have associated intracellular iron accumulation with Parkinson's disease[59], it

would be of considerable interest to explore whether certain features of this neurodegenerative condition, such as the loss of dopaminergic neurons, are associated with the spread of GBM.

In summary, we described the iron commensalism of MES GSCs in mixed GBM tumors and highlighted ferroptosis as a means for targeting the symbiosis between PN and MES subtypes as a potential treatment for aggressive GBM.

## Methods

**Cell culture and reagents**. A panel of glioblastoma stem cells (GSCs) includes four proneural subtypes, 448T, X01, X02, and 528, and three mesenchymal subtypes, 0502, 83, and 1123 were grown in DMEM/F-12 (Corning) supplemented with B27 (Invitrogen), EGF (10 ng/ml, R&D Systems), bFGF (5 ng/ml, R&D Systems), 50 U/ml penicillin, and 50 U/ml streptomycin at 37 °C with 5% $CO_2$ as previously reported [16]. For hypoxic experiments, cells were cultured in Whitley H35 Hypoxystation (HypOxygen) providing 1.0% oxygen and 5% $CO_2$. Reagents were purchased from Selleckchem for RSL3, erastin, and flupenthixol; from Sigma for benserazide, dopamine, ferric chloride, ferristatin II, N-Acetylcystein, and L-Dopa; from Millipore for transferrin, from Uchem for ferrostatin-1 and sorafenib, and from Biovision for PP2. To prepare conditioned media, fifty thousand GSCs in 2 ml media were seeded into 6-well plate. The conditioned media were harvested at 1, 3, 5 days after culturing GSCs for further experiments. Co-culture was performed with Falcon® Permeable Support for 6-well plate (Corning) or by labeled cells with CMTPX (Cayman) and Hoechst 33342 (Sigma).

**Amino acid deprivation media preparation**. Amino acid deficient DMEM/F12 was purchased (USBiological) and reconstituted as manufacturer's instruction. Individual amino acids were purchased (Sigma) and prepared as a 100X stock solution which then used to make the corresponding final concentration in DMEM/F12 full media.

**DRD5 knockdown using siRNA**. PN 528 and MES 83 were seeded at $2 \times 10^5$ cells followed by reserve transfection siRNA control or a pool of three siRNAs targeting DRD5 (Santa Cruz) at a final concentration of 50 nM each for 3 days using X-tremeGENE™ siRNA Transfection Reagent (Roche). Cell pellets were then collected for immunoblot assay and MTS assay.

**Quantitative PCR analysis**. Total RNAs were extracted from cells using TRIzol reagent (Invitrogen) as manufacturer's instruction, and further reverse-transcribed into cDNAs using QPCR RT Master Mix (Toyobo). Quantitative real-time polymerase chain reaction (QPCR) was performed using ABI Prism 7900 HT Sequence Detection System (Applied Biosystems) or QuantStudio 6 Flex (Applied Biosystems). SYBR green real-time PCR master mixes (Life Technologies) were used for triplicates of each PCR reaction. ΔΔCt method was used for gene expression analysis with *18S* as an internal reference gene List of primers used was in Table S1.

**Immunoblot assay**. Cells lysates were prepared using lysis buffer (150 mM NaCl, 1% tritonX-100, 0.5 % sodium deoxycholate, 0.1% SDS, 50 mM tris pH 8) containing protease inhibitor and phosSTOP phosphatase inhibitor cocktail (Roche). Protein concentration was measured using BCA protein assay (Pierce) for western blot analysis. Primary antibodies were purchased from Santa Cruz for transferrin (sc-374441), TfR1 (sc-32272), DRD2 (sc-5303), DRD3 (sc-136170), DRD4 (sc-136169), DRD5 (sc-376088), TH (sc-25269), PAH (sc-271258), and DDC (sc-293287); from Cell Signaling Technology for SOX2 (#3728), CD44 (#3570), pSrc (#6943), Src (#2108), pSTAT3 (#9131), STAT3 (#9139), pERK (#9101 S), ERK (#9102 S), and PARP (#9542 S); from Novus Biologicals for HIF1α (NB100-449) and DRD1 (NBP2-66807); from Abcam for 4-HNE (ab46545), β-actin (ab6276), and HFE (ab133369). Secondary antibodies include HRP conjugated anti-mouse IgG (ab6728) from Abcam and anti-rabbit IgG (G-21234) from Innovative Research. Antibody concentration was 1:1000 for DRD1, DRD2, DRD3, DRD4, DRD5, TH, PAH, and DDC; 1:2000 for transferrin, TfR1, SOX2, CD44, pSrc, Src, pSTAT3, STAT3, pERK, ERK, PARP, HIF1α, and HFE; 1:3000 for 4-HNE; 1:5000 for β-actin and secondary antibodies.

**Analysis of gene expression and patient survival**. Gene expression datasets were downloaded from cBioportal (https://www.cbioportal.org/) for the Cancer Genome Atlas (TCGA, Nature 2008) dataset[60–62] and from Ivy Glioblastoma Atlas Project (Ivy GAP) portal (http://glioblastoma.alleninstitute.org)[63–65]. Gene expression was analyzed using GraphPad Prism 7.0 and visualized as an heatmap or a boxplot. For survival analysis, patients were classified upon gene expression level; higher (equal or greater than 75% quartile), lower (equal or less than 25% quartile), and middle for the rest 50%. For analysis of double genes expression, high or low was determined based on the median expression for each gene. Kaplan–Meier's graphs were then plotted using GraphPad Prism 7.0. A *p*-value of log-rank (Mantel-Cox) test was shown. Prognosis of GBM patients upon *TFRC* expression in combined datasets was plotted on OSgbm (http://bioinfo.henu.edu.cn/GBM/GBMList. jsp)[66–71]. Gene expression in PN and MES from multiple datasets[71–82] was analyzed and downloaded from Gliovis project (gliovis.bioinfo.cnio.es)[83].

**Gene set enrichment analysis (GSEA)**. GBM patients' gene expression data from Ivy GAP dataset was downloaded and classified as PN and MES groups. GSEA was performed upon Affymetrix HT_HG-U133A microarray gene chip and Gene Ontology Biological Process gene set database from GSEA_4.0.3 (https://www.gsea-msigdb.org/gsea/index.jsp)[84,85].

**Cell viability assay**. Appropriate numbers of cells in 1 ml of culture medium were seeded in 12-well plates. After treatment with vehicle, or drugs, or amino acid(s) deprivation media as indicated, cell viability was assessed via MTS assay for measuring absorbance at 490 nm, or by direct cell counting. MTS solution was prepared following the manufacturer's instruction (Promega). For cell counting, cells were stained with trypan blue and counted manually or using Countess II FL (Invitrogen).

**Patient sample collection and preparation**. Fresh tissue samples or paraffin-embedded tissue sections were from Wonju Severance Christian Hospital as approved by the Committee of Institutional Review Board (Approval number: CR320372) with informed consent forms obtained from participants. Fresh glioblastoma tissues, sampled by neurosurgeon from two different anatomical regions of the tumor namely center and margin regions, were fixed with 4% paraformaldehyde at 4 °C for 48 h and then with sucrose 30% at 4 °C until tumors sink to the bottom of vial. The fixed tissues were embedded in optimal cutting temperature (OCT) compound and then cut into 8-µm sections. Sliced samples were then stained for visualizing the expression of proteins of interest.

**Immunohistochemistry and immunofluorescence staining**. Tumor tissues obtained from patients or xenograft mouse models were used for immunohistochemistry or immunofluorescence assay as previously published[9]. Paraffin sections were underwent rehydration and antigen retrieval using citrate buffer (pH 6.0). Then, the overall procedure for both OCT- and paraffin-embedded tissues includes blocking sections with hydrogen peroxide for 30 min. After washing with water, samples were permeabilized by incubating in phosphate-buffered saline with 0.25% Triton X (PBST) for 30 min, followed by blocking in normal goat serum diluted with PBST (1:100) for 30 min. Tissue sections were then incubated with primary antibodies against proteins of interest at 4 °C in a humidified chamber overnight. For immunofluorescence staining, sections were incubated with Alexa Fluor™ 488 anti-mouse or Alexa Fluor™ 594 anti-rabbit secondary antibodies, followed by mounting in VECTASHIELD® PLUS Antifade Mounting Medium with DAPI (Vector). Quantification of fluorescence signal was done using ImageJ (https://imagej.nih.gov/ij/)[86]. For immunohistochemistry experiments, Biotinylated Universal Antibody (Horse Anti-Mouse/Rabbit IgG) and ABC solution from VECTASTAIN Elite ABC HRP Kit (Vector) were consecutively treated to the sections following the manufacturer's instructions. The signal was then detected using ImmPACT DAB Peroxidase (HRP) Substrate (Vector). Finally, the sections were stored with Permanent Mounting Medium (Vector).

**Iron measurement and staining**. Iron concentration was measured using either Quantichrome™ Iron Assay Kit (BioAssay System, 18614-83) or Iron Assay Kit (Abcam, ab83366) following manufacturers' instruction[87]. For measuring intracellular iron, cell pellets were harvested after pre-incubation with ferric ions and washed twice with cold phosphate-buffered saline (PBS), followed by lysis in RIPA buffer. Then, iron levels were determined by the absorbance at 590 nm after incubation in iron detection reagent. Absorbance of a sample blank -supernatant without iron detection reagent- was subtracted to determine supernatant iron. The Abcam Iron Assay Kit was used for determining total iron, ferrous ions, and ferric ions by adding or omitting the iron reducer. After pre-incubating with ferric ions and dopamine, cell lysates were prepared by iron assay buffer provided in the Kit to measure iron as instructed in the protocol. The iron level was normalized by protein concentration. For tissue iron staining, Iron Stain Kit (Prussian Blue Stain) (Abcam, ab150674) was used by following manufacturer's procedure. Frozen tissues were washed with distilled water and stained with iron stain solution for 3 min. Tissue morphology was visualized by provided nuclear fast red solution.

**Dopamine measurement**. Conditioned medium from fifty thousand cells cultured for 3 days was collected for dopamine assay using Dopamine ELISA Kit (CAT#KA1887, Abnova). Following the manufacturer's instruction, samples were processed through extraction, acylation, and enzymatic assay. The extraction step was performed from 10 µl of standards and 300 µl of samples mixed in the extraction plate following by addition of 50 µl of assay buffer and 50 µl of extraction buffer. After shaking incubation for 30 min, the plate was washed twice with washing buffer provided, followed by drying. The acylation step was continued by adding 50 µl of acylation buffer and 25 µl acylation reagent into the extraction plate following 15 min incubation time. The plate was then washed and dried before incubating in 175 µl of hydrochloric acid for 10 min on shaker. The final enzymatic assay was continued with 25 µl of enzyme solution, 25 µl of the extracted standards and 50 µl of the extracted sample following the protocol. Absorbance was measured within 10 min using the microplate reader under the wavelength of 450 nm.

**Cystine uptake assay**. Cystine uptake assay protocol was described previously[88]. In short, GSCs cells was seeded and treated with sorafenib. The next day, cells were starved from cystine for 30 min before exposing to selenocystine (Sigma) for another 30 min. Cells were then incubated in MES buffer (Sigma) containing fluorescein O,O′-diacrylate (FOdA) (Sigma) and tris(2-carboxyethyl)phosphine (TCEP) (Sigma). The fluorescence signal formed between selenocystine and the two probes representing cystine uptake capacity were visualized and fluorescence microscope and quantified using MetaMorph software.

**Cytosolic ROS measurement**. Twenty thousand cells in 1 ml culture medium were seeded into 12-well plate. Two days after seeding, cells were treated with vehicle, dopamine, NAC, and flupenthixol for 24 h. Cell pellets were resuspended in PBS and incubated with 2.5 μM of CM-H2DCFDA (Invitrogen) for 5 min. Green fluorescence signal indicating cytosolic ROS was visualized by fluorescence microscope (Olympus) and quantified using MetaMorph software.

**Lipid peroxidation measurement**. Fifty thousand cells were seeded to glass bottom confocal dish (SPL) in 2 ml culture medium. Two days after seeding, cells were treated with vehicle, dopamine, and RSL3 in the presence of 0.5 μM of BODIPY™ 581/591 C11 (Invitrogen) for 12 h. The red and green fluorescence images, together with the bright-field image, were taken simultaneously under confocal microscope with double wavelength excitation (488 and 568 nm, for green and red, respectively). The ratio of green/red fluorescence signal was determined upon incubation with BODIPY™ 581/591 C11 as previously reported[89]. Twenty thousand cells were seeded in 96-well Costar with clear bottom (Corning) coated by laminin (Sigma) in 100 μl culture media and treated as above following by fluorescence reading with FlexStation II (Molecular Devices).

**Ectopic xenograft tumor models**. Animal experiments were performed under the approval of Institutional Animal Care and Use Committee (IACUC) of Yonsei University Wonju College of Medicine (Approval number: YWC-170907-3). For heterotopic models, five millions of MES 83 cells were inoculated subcutaneously to the right flank of 5-weeks-old female Balb/c nude mice. When tumors were tangible, mice were randomly assigned into four groups (n = 5 for each group). The animals in the four groups were intraperitoneally treated with vehicle, dopamine (25 mg/kg), erastin (10 mg/kg), or dopamine (25 mg/kg) plus erastin (10 mg/kg) every days for 19 days. Tumor dimension and body weight were examined every other day. Tumor volume was determined as equals ½ × (width² × length). Tumor weight was measured on the day of mice sacrificed. For orthotopic models, twenty thousand 83 cells in 2 μl of DMEM/F12 medium were stereotactically transplanted into the right striatum of 5-weeks-old female Balb/c nude mice. The injection point was coordinated on the skull as 1.0 mm to the right of the midline, 1.0 mm posterior to the bregma, and at a depth of 2.5 mm. Mice was then randomly divided to four groups receiving vehicle, L-Dopa (25 mg/kg), sorafenib (30 mg/kg), or L-Dopa (25 mg/kg) with sorafenib (30 mg/kg) treatment. Benserazide (12 mg/kg) was simultaneously administrated with L-Dopa to inhibit peripheral DDC. Drugs were daily administrated intraperitoneally from one week after the transplantation until the end of the experiment. Mice were sacrificed upon severe weight loss or neurologic symptoms. Mouse brains were collected and fixed in 4% paraformaldehyde for paraffin section and immunostaining. Kaplan–Meier plots were used for analysis of mice survival.

**Statistics and reproducibility**. Student's *t*-test was used to compare means of two data groups. One-way ANOVA was used to compare means of two or more data groups. All graphs and statistical analysis were performed using GraphPad Prism version 7.0. Data was reported as mean ± SD (n ≥ 3, except indicated) and p < 0.05 was considered significant. Asterisks refer to *p ≤ 0.05, **p ≤ 0.01, ***p ≤ 0.001, and ****p ≤ 0.0001.

**Reporting summary**. Further information on research design is available in the Nature Research Reporting Summary linked to this article.

## Data availability
Source data is available in Supplementary Fig. 9 and within Supplementary Data. Other data and materials will be available upon request to Y.J.

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

## Acknowledgements

This research has been financially supported by National Research Foundation of Korea (NRF) grant funded by the Korea government (MSIT) (2017R1A5A2015369 and 2020R1A2C1004684 to Y.J.) and by a grant of the Korea Health Technology R&D Project through the Korea Health Industry Development Institute (KHIDI), funded by the Ministry of Health & Welfare (HI18C2196 to Y.J.). We thank Professor Kyu-Sang Park for providing CM-H2DCFDA and Biorender, and Cuong P. Ha for his support with the orthotopic brain tumor model.

## Author contributions

Y.J. supervised the study. Y.J. and V.T.A.V. conceived the hypothesis, designed experiments, and wrote the manuscript. S-H.K. conceived the hypothesis. V.T.A.V., T.N.M.H. performed the experiments with assistance from S-H.K. and J.O.

## Competing interests

The authors declare no competing interest.
