## [Peer Review File · Communications Biology]

Reviewers' comments:

Referee #1:

The manuscript by Vo et al. reports that the PN glioblastoma stem cells (GSCs) secreted both DA and TF, which in turn promoted the proliferation of neighboring MES GSCs, contributing to tumor progression. This work is interesting and its findings provided biological evidence of intra-tumoral commensalism in heterogeneous GBM, and an insightful rationale of targeting ferroptosis for treating GBM. Nevertheless, some issues should be addressed before the acceptance.

1. The protein expression of TF in PN-X02 was hardly detected, however it's highly expressed in CM of PN-X02, please have alternative blot or clarify it.
2. In line 600, the writing of "DA synthesis" was confused, the protein expression of DA synthesis related enzymes was examined.
3. The statistical analysis should be provided in Fig.3e, supplementary Fig. 4a, Fig. 4b and Fig.4g.
4. In line 151-153, the author stated that MES GSCs preferentially uptake MES GSCs preferentially uptake extracellular iron through endocytosis by forming complexes with TfR1 and TF originating from PN CSCs. However, there is no data to support this conclusion.
5. In supplementary Fig. 4g, the 4-HNE staining signal was not observed.
6. It seems DA alone could not enhance ferroptosis. Only combining with cystine deprivation or system Xc- inhibition, DA is pronounced to induce ferroptosis. Additional experiments are required to clarify and discuss.
7. All GSCs exhibited increased intracellular ROS generation upon DA treatment, however DA alone was not able to promote ferroptosis. Please clarify it.
8. In fig.4e, without DA, Erastin was not able to induce ferroptosis. However, 4-HNE increase was observed in erastin group. please discuss it.
9. The scheme Fig.4h should be re-drawn, since it is oversimplified and clear.

Referee #2:

In this research article, the authors aimed to explore the mechanism through which GBM heterogeneity is coordinated to promote tumor progression. And they found that the proliferation and susceptibility toward ferroptosis of MES GSCs could be induced by dopamine and transferrin which are secreted by PN GSCs. To reveal the mechanism of a commensal relationship between PN and MES GSCs, the authors have done considerable work both in vivo and in vitro. However, several issues are needed to be addressed before further consideration.

1. The authors demonstrated that TfR1 expression was higher in the central MES region. However, in figure 1E, it seems that the expression of TfR1 in the margin part is higher than center part. Please confirm that the conclusion is consistent with the result.
2. The authors demonstrated that dopamine enhances ferroptosis of GBMs and sorafenib could mediate system Xc inhibition. And they proved that combined treatment with sorafenib and DA caused a further increase in overall survival compared with sorafenib alone. As system Xc plays important role in ferroptosis and sorafenib could induce ferroptosis by inhibiting it, please prove that system Xc was truly inhibited by sorafenib.
3. GPX4 is a crucial protein in ferroptosis. Whether combine treatment with sorafenib and dopamine would influence its' expression level when compared with treatment with sorafenib or dopamine alone?
4. Whether the susceptibility to ferroptosis of MES GSCs enhanced by dopamine could be reversed by ferroptosis inhibitor?

Referee #3:

In this current manuscript, the authors are trying to address essential issues pretending to develop effective therapies against GBM, which is how the molecular heterogeneity promotes oncogenic progression. The authors here demonstrated that proneural GSCs could secrete dopamine and transferrin, which induced the proliferation of mesenchymal GSC and promoted gliomagenesis. This manuscript has several strengths, including investigating a critical issue in aggressive disease, using the patient-derived GSC model, and appropriate interpretation of the data. However, the manuscript also has some following significant weaknesses.

1. As the author indicated, GBM is highly heterogeneous, with cells that can transit between different molecular states (PMID:24925914). Moreover, a recent report demonstrated that GBM

cells could be in a hybrid state and acquire other states based on microenvironmental cues. One of the issues with this manuscript's experimental setup is the investigator utilizing bulk RNA-seq data that ignores the heterogeneity at the single-cell level while trying to study the intertumoral heterogeneity. It will be essential to investigate intertumoral heterogeneity because many different subtypes can be present in the same tumor. One such experiment should be carried out where authors co-culture genetically labeled (GFP or RFP) different subtypes of GSC and investigate how such interaction can influence the growth of different subtypes of GBM.

2. A global pathway enrichment analysis is necessary for genes enriched in different subtypes. These resources are open source and should be implemented to evaluate differentially regulated genes (Suppl. Fig. 1C and D) and their relation to other pathways.

3. The discrepancy in DRD1 mRNA expression clinically (Suppl Fig 1D) and experimentally (Fig 1D) is troubling. Authors should consider using other public sources of GBM patient data, such as <https://www.proteinatlas.org> and <http://gliovis.bioinfo.cnio.es>, to validate their findings.

Moreover, it would be critical to investigate all the DRD receptors in patient's GBM in Fig 1e.

4. Validating the finding in Fig 1E in multiple GBM tissues would be critical.

5. Fig. 1F, the IF picture should be captured in higher magnification. It would be essential to capture images in multiple areas digitally analyze by ImageJ each picture for expression. From the current images presented in the manuscript, it is unclear how these images represent the total disease area.

6. Line 126-127, MES originated in hypoxia, and please add a reference that supports this statement.

7. It is unclear if the TF and Fe concentration used in the experimental setting is physiologically relevant. Why did the authors choose to use this concentration? Would you please justify using references?

8. According to Fig 2f, TF and Fe can stabilize Hif selectively in the MES lines, and authors claim that such stability may be responsible for increased growth observed in Fig 2a. However, it is not explained by the authors why we also kept increased growth in the normoxic conditions in Fig 2A as both TF and Fe didn't alter the HIF expression in Fig 2a. is it possible that increased growth HIF independent?

9. As the authors mentioned, there are conflicting reports published in the literature regarding the specific DRD receptors' involvement in gliomagenesis. Some reports indicated that DRD3-4 is critical (PMID:32358191). Others indicated that maybe DRD2 is critical (PMID:30651332). Based on these conflicting results and the lack of specificity for some of the DRD inhibitors, it would be essential to demonstrate the importance of DRD5 by genetically knockdown the receptor and perform the same experiments in fig 3.

10. Please provide a scientific rationale for the use of ferroptosis induced in fig 4. Is this therapeutic approach will work in combination with other therapies?

11. Fig 4a, the therapeutic effect is not selective to MES; the viability of the proneural GSC is significantly reduced in the Veh condition. Why?

12. The Subcutaneous tumor model in Fig 4e does not provide any added information for developing therapy in GBM. I would recommend removing this fig in the supplementary fig.

13. Fig 4f should be separated by PN and MES subtypes as this manuscript's main hypothesis address the molecular heterogeneity of the GBM.

EDITOR'S COMMENTS

Your manuscript entitled "Iron commensalism of mesenchymal glioblastoma promotes ferroptosis susceptibility upon dopamine treatment" has now been seen by editors, whose comments are appended below. You will see from their comments copied below that while they find your work of potential interest, they have raised quite substantial concerns that must be addressed. In light of these comments, we cannot accept the manuscript for publication, but would be interested in considering a revised version that addresses these serious concerns.

Thank you for your time and consideration on our work.

We hope you will find the referees' comments useful as you decide how to proceed. Should further experimental data or analysis allow you to address these criticisms, we would be happy to look at a substantially revised manuscript. However, please bear in mind that we will be reluctant to approach the referees again in the absence of major revisions.

In particular, please note that the following revisions would be necessary for us to contact our referees again:

The authors reported that PN-GSCs can secrete DA and TF to facilitate GBM progression. DA supports PN-GSCs growth in an autocrine manner via DA receptor D1, while DA also stimulates MES-GSCs proliferation in a paracrine and iron-dependent manner via upregulating TF receptor 1. By in vitro/vivo experiments and clinical correlative analyses, they indicate that DA- and TF-mediated iron commensalism between PN-GSCs and MES-GSCs may lead to poor survival, while ferroptosis inducers may target these features and show therapeutic advantages.

The overall judgement is that this work exhibits some novelties and potential clinical applications. But a substantial amount of experimental validations and mechanistic explorations are still needed to support the major conclusions.

We thank you and the reviewers for all valuable comments and would like to address them point-by-point as below.

In light of the comments, we have added more data and changed the structure of the revised manuscript without any changing in the flow of the story. Thus, all the figures quoted in our answers below are in the order of the revised manuscript.

1. It is known that DA may promote glioblastoma progression (10.1073/pnas.1920154117). Similar mechanisms have been discovered in macrophages (10.1016/j.bcp.2017.12.001). This study emphasized the role of DA on MSC-GSCs. But it may also promote tumor progression via acting on tumor-associated macrophages.

We appreciate your insightful question. It has been reported that DA may promote glioblastoma progression suggesting potential benefits of treating glioblastoma by combining DA receptor antagonist trifluoperazine with radiation¹. Another report also implied functions of DA by proving that glioblastoma could synthesize and secrete DA and utilized DA in an autocrine manner².

Our study attempted to clarify that the GSCs capable of synthesizing and secreting DA would be of proneural (PN) subtype (Fig. 1e). DA secreted from PN would be used in an autocrine manner by the PN cells and at the same time promoted iron uptake into mesenchymal (MES) cells (Fig. 4d) in the heterogenous tumor microenvironment of glioblastoma. As you pointed out, the same phenomenon that DA increased iron uptake leading to increase of cellular stress response has been proven in macrophages³. Thus, it is reasonable to wonder if DA may affect tumor-associated macrophages (TAMs) via which alter the characteristics and ferroptosis sensitivity of glioblastoma cells.

To validate this rationale, we co-cultured GBM cells with TAMs namely U937 or THP-1 using permeable membrane allowing cell-cell communication ([https://ecatalog.corning.com/life-sciences/b2c/US/en/Permeable-Supports/Inserts/Falcon% C2% AE-Cell-Culture-Inserts/p/353493](https://ecatalog.corning.com/life-sciences/b2c/US/en/Permeable-Supports/Inserts/Falcon%C2%AE-Cell-Culture-Inserts/p/353493)). U937 and THP-1 were widely used in the previous reports investigating functions of TAMs in gliomas⁴⁻⁷.

TAMs were differentiated and co-cultured with glioblastoma cells following protocol described previously^{7,8}. Differentiation of TAMs using phorbol-12-myristate 13-acetate (PMA) was confirmed by expression of CD36⁹ (Editor's fig. 1a), followed by co-culturing with GSCs (TAMs were in upper compartment). In agreement with other reports^{4,7}, presence of TAMs promoted the growth of GSCs, especially MES cells (Editor's fig. 1b), suggesting that targeting both TAMs and tumor cells would be beneficial in glioblastoma treatment. It is of interest to note that TAMs show substantial sensitivity to ferroptosis inducer RSL3 and are rescued by ferroptosis inhibitor ferrostatin-1 while DA had no effect on their growth (Editor's fig. 2a). In addition, even if being co-cultured with U937 or THP-1, respectively, both PN and MES cells remained similar response to ferroptosis inducers and DA treatment as in single culture, still promoting ferroptosis (Editor's fig. 2b, c). While strategies suppressing TAMs to target GSCs remained poorly productive¹⁰, these data might indicate ferroptosis inducer as a promising therapy that target both GSCs and TAMs.

Taken together, **DA mediated GSC growth would be independent of TAM-mediated GSC growth.**

Editor's figure 1. TAMs promoted GSCs cell growth. (a) TAMs were successfully differentiated using 100 nM of PMA. (b) MTS assay for cell viability of GSCs in co-culture with TAMs.

Editor's figure 2. Effects of DA and ferroptosis inducer RSL3 on TAMs and GSCs co-cultured with TAMs. (a) TAMs were susceptible to ferroptosis inducers but not DA alone. (b, c) Co-culture with TAMs THP-1 (b) and U937 (c), did not affect the response of GSCs to DA and/or ferroptosis inducer.

2. With current evidence, it remains elusive whether PN-GSCs are the major source of DA and TF in tumor tissues. Do PN-GSCs constitutively produce DA and TF in situ. If not, which intra-tumoral factors can stimulate the secretion of DA and TF from PN-GSCs? Anatomical expression profile data is only indicative, while *in vitro* data may not represent *in vivo* situations.

We appreciate your comment. While it is difficult to fully address the question, we here try to explain using multiple approaches: integrating the knowledge from literature, investigating gene expression data, and checking the secretion of DA and TF from PN-GSCs upon to identify potential factor(s). Although not able to provide in situ tumor data for DA and TF secretion, due to technical limitation of generating a xenograft mouse model of PN GSCs^{11,12}, we further intensively analyzed anatomical expression dataset from IVY-GAP database available in public, and followed by *in vitro* experiment.

Secretion of DA

DA secretion from glioblastoma cells was previously reported by measuring DA secretion and showing expression of DA synthesis enzyme tyrosine hydroxylase (TH) in cell culture condition². In our study, using the same approach, we classified that TH, and other DA synthesis enzymes were highly expressed in PN-GSCs and conditioned media from PN-GSCs had elevated DA content (Fig. 1e).

The secretion of DA from intracellular vesicles is tightly regulated by the formation of SNARE complex including vesicular SNARE Synaptobrevin-2/VAMP-2 and Syntaxin-1 (STX1)/SNAP-25. DA is released from typical regions called active zones in the plasma membrane¹³. When calcium channels, being tethered to active zones by Regulating Synaptic Membrane Exocytosis (RIMS) proteins^{14,15}, opens, it allows calcium to enter and quickly cause vesicle fusion through calcium sensor synaptotagmin (SYTs) family of proteins¹³. We firstly investigated IVY-GAP database and found an exclusive expression of all the genes involved in DA release in the PN region of the tumor tissues (Editor's fig. 3). By carrying out further gene set enrichment analysis (GSEA) using the same dataset suggested by reviewer #3, we found dopamine secretion pathway was highly enriched in PN but not in MES (Editor's fig. 4, as *Supplementary Fig. 1e (left) with description of the result in lines 106-108 and the method in lines 375-379 of the revised manuscript*). Given that calcium could be one of the crucial environmental factors stimulating DA secretion¹³⁻¹⁷ in both synapses and cancer cells^{16,17}, we depleted calcium from the culture media of PN-GSCs and checked for DA level by ELISA. Calcium chelator EGTA suppressed DA release from the PN-GSCs (Editor's fig. 5). These data suggest that DA secretion from the PN-GSCs may be stimulated by calcium.

Editor's figure 3. Expression of DA secretion regulating genes in IVY-GAP database. DA secretion genes were highly expressed in PN, but not MES regions of glioblastoma.

Editor's figure 4. Gene set enrichment analysis (GSEA) of IVY-GAP database. Dopamine secretion pathway was enriched in PN compared to MES.

Editor's figure 5. Effect of calcium chelator on secretion of DA in PN-GSCs. EGTA (2 μ M) treatment reduced DA secretion from PN 528 cells.

Secretion of TF

A previous study reported that only GSCs and hepatoma among various cancers had TF-specific enhancer in their genome¹⁸, following to prove that TF could be synthesized and secreted from glioblastoma specimens.

We here showed that mRNA expression of TF was higher in PN compared to MES-GSCs and that PN-GSCs secreted TF into the conditioned media (Fig. 1c, d). Considering that transferrin is an iron-carrying

factor¹⁹, we rationalized iron concentration would be one of the environmental factors regulating TF secretion. Thus, by exposing PN-GSCs in long-term (3 days) with different iron concentrations, we interestingly found a high concentration of environmental iron (from 100 μM) reduced TF secretion from PN-GSCs (Editor's fig. 6).

Editor's figure 6. Effect of iron on TF. Exposure to high concentration of iron reduced TF in cell lysis and media of PN 528 cells.

Relevant data and discussion on the secretion of TF and DA will be included as Supplementary Fig. 1 g, h and in lines 113-119 of the revised manuscript.

3. Mechanistic explorations of how DA and TF enhance MES GSC proliferation should be deeper.

Please be noted again that all the figures quoted below are in the order of the revised manuscript.

We appreciate your comment. We assume this would be closely relevant to the question 8 of reviewer #3.

In this study, we have proven that DA and TF were both secreted from PN-GSCs. In which, DA maintained PN growth in an autocrine manner through DRD5 and at the same time, promote iron uptake in MES by increasing iron importer TfR1 (Fig. 4d) while TF was another undeniable factor in iron trafficking.

Iron is important factor promoting glioblastoma growth^{18,20} and strategies to target iron to cure glioblastoma have been proposed²¹. Our study further identified that of two important subtypes, MES-GSCs relied more on iron for cell growth (Fig. 3a, b, f) and that DA and TF, by promoting iron uptake,

may (1), enhance MES-GSCs proliferation, and (2), make MES-GSCs become susceptible to iron-dependent cell death ferroptosis.

Iron signaling had potentially reciprocal links to Src pathway^{22,23}. We also showed that, the downstream factor of iron signaling in MES would be oncogenic Src (Fig. 3c, f) and its downstream ERK but not STAT3 (Supplementary fig. 3b). While DA treatment alone did not increase phosphorylation of Src in MES, we performed other experiment here with co-treatment of DA and iron. DA seemed to increase more pSrc compared to single treatment of iron (Editor's fig. 7).

In hypoxia condition, Src-ERK signaling was important for maintaining hypoxia-inducible factor Hif1 α ²⁴⁻²⁶. Consistently, we observed iron dependent Hif1 α stability through Src activation upon TF or iron treatment in MES (Fig. 3f). Thus, DA and TF enhanced iron uptake and iron promoted MES proliferation potentially through Src-ERK pathway.

To deepen and clarify this mechanism, we treated MES GSCs with two different types of Src inhibitors SU6656 and PP2. Among the two inhibitors, PP2 suppressed pERK while SU6656 only inhibited pSTAT3 (Editor's fig. 8). Consistently, PP2 but not SU6656 was able to abolish Hif1 α expression stabilized by TF or iron treatment under hypoxia (Editor's fig. 9, as *Supplementary Fig. 3d with description of the result in lines 170-172 of the revised manuscript*), emphasizing the specificity of Src-ERK-Hif1 α axis activated by iron. The improvement of MES proliferation by TF or iron treatment in both normoxia and hypoxia was also abolished by PP2 (Editor's fig. 10, as *Fig. 3g with description of result in lines 172-173 of the revised manuscript*).

Editor's figure 7. Activation of Src upon iron and DA treatment. Western blot showed that co-treatment of DA and iron further enhanced pSrc.

Editor's figure 8. Inhibition of Src by SU6656 or PP2. Treatment of SU6656 (5 μM) or PP2 (20 μM) differently suppressed Src downstreams pSTAT3 and pERK, respectively.

Editor's figure 9. Regulation of iron-Src-Erk signaling pathway by Src inhibitor. PP2 (right), but not SU6656 (left) inhibited pERK thus reduced Hif1α stabilized by TF or iron treatment.

Editor's figure 10. MES GSCs cell growth upon Src inhibitor treatment. PP2 abolished TF- or iron-induced cell viability both in normoxia and hypoxia.

4. Although sorafenib was reported to trigger ferroptosis in HCC, it fails to trigger ferroptosis across a wide range of cancer cell lines (10.1038/s41419-021-03998-w).

We appreciate your concern. In the study by *Zheng et al.*, multiple cell lines have been tested for sensitivity upon sorafenib treatment, including one glioma cells U-373. However, U-373 turns out to be resistant to classical ferroptosis inducer erastin, and it may explain why sorafenib also failed to induce ferroptosis on this cell line²⁷.

Sorafenib has been confirmed to induce ferroptosis in hepatocellular carcinoma (HCC)^{28,29} following the observation that sorafenib can inhibit system Xc^{30,31} confirmed by decrease of glutamate release and glutathione level³⁰. Besides HCC, sorafenib was shown to exert ferroptosis in other solid cancer cells, such as kidney cancer or lung cancer with a positive correlation of ferroptotic potency to erastin that could be reverse by deferoxamine^{32,33}.

Glioma cells U87MG and F98 were validated to undergo ferroptosis by flow cytometry upon sorafenib treatment³⁴. Sorafenib has been studied previously as potential for glioma treatment due to its ability to penetrate blood-brain barrier³⁵, thus suitable for the stereotactic brain tumor model.

To address ferroptotic susceptibility of GBM to sorafenib, we further carried out *in vitro* ferroptosis assays using GSCs for sorafenib (Editor's fig. 11). In the assay, cell death was clearly rescued by ferroptosis inhibitor Fer-1 suggesting the sorafenib-induced ferroptosis of the GSC while DA treatment also synergized with sorafenib to promote cell death (Editor's fig. 12). Consistently, in animal model using MES 83 GSCs, DA plus sorafenib further enhanced ferroptosis in the tumor tissues indicating by 4-HNE elevated signal (Fig. 5d). In addition, we performed an assay if sorafenib functionally inhibits system Xc by measuring cystine uptake upon sorafenib treatment. The data clearly show that sorafenib treatment suppressed cystine uptake to GSCs (please refer to the rebuttal for the reviewer #2, question 2).

Taken together, sorafenib was a proper ferroptosis inducer for the cancer cells tested in this manuscript.

Editor’s figure 11. GSCs cell viability upon sorafenib treatment in a dose-dependent manner. MTS assay showed that sorafenib exerted growth inhibitory effect in all GSCs line tested.

Editor’s figure 12. Ferroptosis induction by sorafenib treatment. MTS assay showed that sorafenib induced ferroptosis in MES 83.

5. In several figures, statistical analyses are missing. Some figures seem to contradict conclusions.

We appreciate your comments.

The statistical significance will be provided in Fig. 4e, Supplementary Fig. 5a, Fig. 5b and Fig. 7 as requested by reviewer #2.

Regarding the questions on conflicts between figures and conclusions, we have answered for reviewers in the rebuttal files (question 1, 2, 5, 8 from reviewer #1; and question 1 of reviewer #2).

References

- 1 Bhat, K. *et al.* The dopamine receptor antagonist trifluoperazine prevents phenotype conversion and improves survival in mouse models of glioblastoma. *Proc Natl Acad Sci U S A* **117**, 11085-11096, doi:10.1073/pnas.1920154117 (2020).
- 2 Caragher, S. P. *et al.* Activation of Dopamine Receptor 2 Prompts Transcriptomic and Metabolic Plasticity in Glioblastoma. *J Neurosci* **39**, 1982-1993, doi:10.1523/JNEUROSCI.1589-18.2018 (2019).
- 3 Dichtl, S. *et al.* Dopamine promotes cellular iron accumulation and oxidative stress responses in macrophages. *Biochem Pharmacol* **148**, 193-201, doi:10.1016/j.bcp.2017.12.001 (2018).
- 4 De Boeck, A. *et al.* Glioma-derived IL-33 orchestrates an inflammatory brain tumor microenvironment that accelerates glioma progression. *Nat Commun* **11**, 4997, doi:10.1038/s41467-020-18569-4 (2020).
- 5 Bao, L. & Li, X. MicroRNA-32 targeting PTEN enhances M2 macrophage polarization in the glioma microenvironment and further promotes the progression of glioma. *Mol Cell Biochem* **460**, 67-79, doi:10.1007/s11010-019-03571-2 (2019).
- 6 Wang, Y. *et al.* Hypoxia and macrophages promote glioblastoma invasion by the CCL4-CCR5 axis. *Oncol Rep* **36**, 3522-3528, doi:10.3892/or.2016.5171 (2016).
- 7 Zhou, W. *et al.* Periostin secreted by glioblastoma stem cells recruits M2 tumour-associated macrophages and promotes malignant growth. *Nat Cell Biol* **17**, 170-182, doi:10.1038/ncb3090 (2015).

- 8 Coniglio, S., Miller, I., Symons, M. & Segall, J. E. Coculture Assays to Study Macrophage and Microglia Stimulation of Glioblastoma Invasion. *J Vis Exp*, doi:10.3791/53990 (2016).
- 9 Alessio, M. *et al.* Synthesis, processing, and intracellular transport of CD36 during monocytic differentiation. *J Biol Chem* **271**, 1770-1775, doi:10.1074/jbc.271.3.1770 (1996).
- 10 Akins, E. A., Aghi, M. K. & Kumar, S. Incorporating Tumor-Associated Macrophages into Engineered Models of Glioma. *iScience* **23**, 101770, doi:10.1016/j.isci.2020.101770 (2020).
- 11 Akter, F. *et al.* Pre-clinical tumor models of primary brain tumors: Challenges and opportunities. *Biochim Biophys Acta Rev Cancer* **1875**, 188458, doi:10.1016/j.bbcan.2020.188458 (2021).
- 12 Haddad, A. F. *et al.* Mouse models of glioblastoma for the evaluation of novel therapeutic strategies. *Neurooncol Adv* **3**, vdab100, doi:10.1093/oaajnl/vdab100 (2021).
- 13 Liu, C. & Kaeser, P. S. Mechanisms and regulation of dopamine release. *Curr Opin Neurobiol* **57**, 46-53, doi:10.1016/j.conb.2019.01.001 (2019).
- 14 Kaeser, P. S., Deng, L., Fan, M. & Sudhof, T. C. RIM genes differentially contribute to organizing presynaptic release sites. *Proc Natl Acad Sci U S A* **109**, 11830-11835, doi:10.1073/pnas.1209318109 (2012).
- 15 Robinson, B. G. *et al.* RIM is essential for stimulated but not spontaneous somatodendritic dopamine release in the midbrain. *Elife* **8**, doi:10.7554/eLife.47972 (2019).
- 16 Bieger, S., Morinville, A. & Maysinger, D. Bisperoxovanadium complex promotes dopamine exocytosis in PC12 cells. *Neurochem Int* **40**, 307-314, doi:10.1016/s0197-0186(01)00093-6 (2002).
- 17 Jeon, H. J. *et al.* Dopamine release in PC12 cells is mediated by Ca(2+)-dependent production of ceramide via sphingomyelin pathway. *J Neurochem* **95**, 811-820, doi:10.1111/j.1471-4159.2005.03403.x (2005).
- 18 Schonberg, D. L. *et al.* Preferential Iron Trafficking Characterizes Glioblastoma Stem-like Cells. *Cancer Cell* **28**, 441-455, doi:10.1016/j.ccell.2015.09.002 (2015).

- 19 Patel, M. & Ramavaram, D. V. Non transferrin bound iron: nature, manifestations and analytical approaches for estimation. *Indian J Clin Biochem* **27**, 322-332, doi:10.1007/s12291-012-0250-7 (2012).
- 20 Ferrarelli, L. K. Iron fuels glioblastoma growth. *Science Signaling* **8**, ec311-ec311, doi:doi:10.1126/scisignal.aad7099 (2015).
- 21 Behr, S. C. *et al.* Targeting iron metabolism in high-grade glioma with 68Ga-citrate PET/MR. *JCI Insight* **3**, doi:10.1172/jci.insight.93999 (2018).
- 22 Cao, H., Chen, J., Krueger, E. W. & McNiven, M. A. SRC-mediated phosphorylation of dynamin and cortactin regulates the "constitutive" endocytosis of transferrin. *Mol Cell Biol* **30**, 781-792, doi:10.1128/MCB.00330-09 (2010).
- 23 Jian, J., Yang, Q. & Huang, X. Src regulates Tyr(20) phosphorylation of transferrin receptor-1 and potentiates breast cancer cell survival. *J Biol Chem* **286**, 35708-35715, doi:10.1074/jbc.M111.271585 (2011).
- 24 Lamberti, M. J. *et al.* Transcriptional activation of HIF-1 by a ROS-ERK axis underlies the resistance to photodynamic therapy. *PLoS One* **12**, e0177801, doi:10.1371/journal.pone.0177801 (2017).
- 25 Lee, H. Y. *et al.* Src activates HIF-1alpha not through direct phosphorylation of HIF-1alpha specific prolyl-4 hydroxylase 2 but through activation of the NADPH oxidase/Rac pathway. *Carcinogenesis* **32**, 703-712, doi:10.1093/carcin/bgr034 (2011).
- 26 Jung, F. *et al.* Hypoxic induction of the hypoxia-inducible factor is mediated via the adaptor protein Shc in endothelial cells. *Circ Res* **91**, 38-45, doi:10.1161/01.res.0000024412.24491.ca (2002).
- 27 Zheng, J. *et al.* Sorafenib fails to trigger ferroptosis across a wide range of cancer cell lines. *Cell Death Dis* **12**, 698, doi:10.1038/s41419-021-03998-w (2021).

- 28 Wang, Q. *et al.* GSTZ1 sensitizes hepatocellular carcinoma cells to sorafenib-induced ferroptosis via inhibition of NRF2/GPX4 axis. *Cell Death Dis* **12**, 426, doi:10.1038/s41419-021-03718-4 (2021).
- 29 Louandre, C. *et al.* Iron-dependent cell death of hepatocellular carcinoma cells exposed to sorafenib. *Int J Cancer* **133**, 1732-1742, doi:10.1002/ijc.28159 (2013).
- 30 Dixon, S. J. *et al.* Pharmacological inhibition of cystine-glutamate exchange induces endoplasmic reticulum stress and ferroptosis. *Elife* **3**, e02523, doi:10.7554/eLife.02523 (2014).
- 31 Dixon, S. J. *et al.* Ferroptosis: an iron-dependent form of nonapoptotic cell death. *Cell* **149**, 1060-1072, doi:10.1016/j.cell.2012.03.042 (2012).
- 32 Lachaier, E. *et al.* Sorafenib induces ferroptosis in human cancer cell lines originating from different solid tumors. *Anticancer Res* **34**, 6417-6422 (2014).
- 33 Li, Y. *et al.* Erastin/sorafenib induces cisplatin-resistant non-small cell lung cancer cell ferroptosis through inhibition of the Nrf2/xCT pathway. *Oncol Lett* **19**, 323-333, doi:10.3892/ol.2019.11066 (2020).
- 34 Chen, D., Eyupoglu, I. Y. & Savaskan, N. Ferroptosis and Cell Death Analysis by Flow Cytometry. *Methods Mol Biol* **1601**, 71-77, doi:10.1007/978-1-4939-6960-9_6 (2017).
- 35 Siegelin, M. D., Raskett, C. M., Gilbert, C. A., Ross, A. H. & Altieri, D. C. Sorafenib exerts anti-glioma activity in vitro and in vivo. *Neurosci Lett* **478**, 165-170, doi:10.1016/j.neulet.2010.05.009 (2010).

REVIEWERS' COMMENTS

Reviewer #1

The manuscript by Vo et al. reports that the PN glioblastoma stem cells (GSCs) secreted both DA and TF, which in turn promoted the proliferation of neighboring MES GSCs, contributing to tumor progression. This work is interesting and its findings provided biological evidence of intra-tumoral commensalism in heterogeneous GBM, and an insightful rationale of targeting ferroptosis for treating GBM. Nevertheless, some issues should be addressed before the acceptance.

Thank you very much for your comments. We would like to answer your concerns point-by-point as below.

In light of the comments from reviewers, we have added more data and changed the structure of the revised manuscript without any changing in the flow of the story. Thus, all the figures quoted in our answers below are in the order of the revised manuscript.

1. The protein expression of TF in PN-X02 was hardly detected, however it's highly expressed in CM of PN-X02, please have alternative blot or clarify it.

We appreciate your point. As suggested, we here provide an alternative blot (Rebuttal fig. 1, as Fig. 1c in revised manuscript).

Rebuttal figure 1. Western blot for TF expression in cell lysis of GSCs panel

2. In line 600, the writing of “DA synthesis” was confused, the protein expression of DA synthesis related enzymes was examined.

We appreciate your point. As you suggested, we clarify the writing of “DA synthesis” to “Expression of DA synthesis related enzymes” in *lines 702-703 of the revised manuscript*.

3. The statistical analysis should be provided in Fig.3e, supplementary Fig. 4a, Fig. 4b and Fig.4g.

Thank you for your point. We put the statistical analysis in all the figures in the revised manuscript.

4. In line 151-153, the author stated that MES GSCs preferentially uptake extracellular iron through endocytosis by forming complexes with TfR1 and TF originating from PN CSCs. However, there is no data to support this conclusion.

We appreciate your concern. Here, we would like to change the writing of “MES GSCs preferentially uptake extracellular iron through endocytosis by forming complexes with TfR1 and TF originating from PN GSCs” to “PN-derived TF may be used preferentially by MES GSCs” in *lines 173-174 of the revised manuscript*.

5. In supplementary Fig. 4g, the 4-HNE staining signal was not observed.

We appreciate your comment. We provide a new experimental blot to replace the old one. The data show that combined treatment of DA and erastin increases 4-HNE in the tumor (Rebuttal fig. 2, as Supplementary Fig. 7c (upper) in the revised manuscript).

Rebuttal figure 2. Western blot for 4-HNE expression in ectopic xenograft tumors upon different treatments including single treatment of DA, erastin, or combined DA and erastin.

6. It seems DA alone could not enhance ferroptosis. Only combining with cystine deprivation or system Xc- inhibition, DA is pronounced to induce ferroptosis. Additional experiments are required to clarify and discuss.

7. All GSCs exhibited increased intracellular ROS generation upon DA treatment, however DA alone was not able to promote ferroptosis. Please clarify it.

We appreciate your valuable questions (#6 and #7). We provide here an integrated answer for those two questions.

We agree with reviewer 1's points showing that DA treatment does not induce ferroptosis by itself even with increased intracellular ROS (Fig. 5b), but is able to promote ferroptosis only in combination with ferroptosis inducers. This is consistent with multiple previous reports as follows.

As intracellular ROS becomes harmful in cellular viability, the cells develop enzymatic ROS scavenging system to overcome lethal intracellular ROS stress¹. One of the hallmarks for ferroptosis includes lipid peroxidation^{2,3} process in which lipid peroxide is elevated under the **accumulation of ROS**, typically **together with the disruption of the lipid peroxidation controlling system** including cystine-glutathione-glutathione peroxidase 4 (GPX4) axis¹. Thus, cytosolic ROS accumulation should be combined with functional defect of intracellular anti-oxidant system (e.g., cystine deprivation, system Xc- inhibitor erastin, or GPX4 inhibitor RSL3) to fully induce ferroptosis.

As reported in the literature that a direct treatment of ROS alone, like H₂O₂, could cause non-ferroptotic cell death not being rescued by classical ferroptosis inhibitor², DA-induced cytosolic ROS in our study is necessary but not sufficient to induce lipid peroxide accumulation and ferroptosis. Indeed, using a specific marker, C11-BODIPY, we confirmed that lipid peroxide was only elevated in GSCs when treated with both DA and RSL3, but not DA alone (Fig. 5c, Supplementary Fig. 5d).

Our data show that DA treatment induces no cell death of GSCs while increases intracellular ROS generation (Rebuttal fig. 3, Fig 5b).

We conducted another experiment treating H₂O₂ directly from 50 to 1000 μM to GSCs. The data confirmed that, low doses of H₂O₂ did not affect GSCs growth, while high doses (~500 μM) started to cause a cell death which could not be rescued by Fer-1 treatment (Rebuttal fig. 4), thus not ferroptosis.

On the other hand, the combined treatment of RSL3, a GPX4 inhibitor, with H₂O₂ enhances ferroptosis compared to the one of RSL3 alone, specifically in the ferroptosis resistant MES-GSCs (Rebuttal fig. 5).

High doses of H₂O₂ consistently exerted severe non-ferroptotic cell death. Consistently, 4-HNE level is also elevated only when H₂O₂ was treated in combination with RSL3, but not H₂O₂ alone (Rebuttal fig. 6).

Rebuttal figure 3. MTS assay for GSCs cell viability upon DA treatment. DA did not show significant effects on GSCs growth.

Rebuttal figure 4. MTS assay for cell viability under treatment of H₂O₂ to the GSCs. H₂O₂ caused cell death which could not be rescued by Fer-1.

Rebuttal figure 5. MTS assay for cell viability under treatment of H₂O₂ in combination with RSL3 in GSCs. H₂O₂ profound ferroptosis with RSL3 treatment.

Rebuttal figure 6. Western blot for 4-HNE expression upon treatment of RSL3 and/or H₂O₂. H₂O₂ elevated 4-HNE in combination with RSL3.

8. In fig. 4e, without DA, Erastin was not able to induce ferroptosis. However, 4-HNE increase was observed in erastin group. please discuss it.

Please be noted again that Fig. 4e has been changed to Supplementary fig. 7c of the revised manuscript.

We appreciate your comment.

In the ectopic xenograft model, we used MES 83 cells and treated with DA, erastin alone or DA and erastin. Combined treatment could elevate ferroptosis in cancer cells leading to reduction of tumor volume and tumor weigh indicating by strong accumulation of 4-HNE. This did not exclude the fact that erastin alone, as system Xc inhibitor, could accumulate a **modest level of 4-HNE** which might not be enough to show meaningful biological consequence. The accumulation of 4-HNE was highly elevated when erastin was treated together with DA. In other word, MES 83 became more sensitive to erastin in the presence of DA.

Some previous studies have reported the similar phenomenon. For instance, *Yang et al.* proposed that knockout of Nedd4 sensitized melanoma cells to erastin treatment. They also checked 4-HNE in the

animal tumor and found that while erastin induced a modest level of 4-HNE, combining with shNedd4 considerably increased 4-HNE more (Yang et al, Nat Comm 2020, Fig. 7e)⁴. In 2021, Zhang et al. showed in a PDX tumor model that erastin could induce 4-HNE, however, the combined of erastin with AZD-8055 remarkably escalated ferroptosis observed through stronger 4-HNE signal (Zhang et al, Nat Comm 2021, Fig. 4i)⁵. Our story proposed another combination of erastin and DA. And the data of 4-HNE in Supplementary fig. 7b, c was in accordance with the overarching idea of the story that treatment of DA sensitizes or enhances MES-GSCs to ferroptosis inducer.

We clarified the expression of 4-HNE in *lines 240-241 of the revised manuscript*.

9. The scheme Fig.4h should be re-drawn, since it is oversimplified and clear.

We appreciate your suggestion and provide. We would like to provide another scheme figure as below (Rebuttal fig. 7, as *Fig. 6 in revised manuscript*).

Rebuttal figure 7.
Hypothetical model

Reviewer #2

In this research article, the authors aimed to explore the mechanism through which GBM heterogeneity is coordinated to promote tumor progression. And they found that the proliferation and susceptibility toward ferroptosis of MES GSCs could be induced by dopamine and transferrin which are secreted by PN GSCs. To reveal the mechanism of a commensal relationship between PN and MES GSCs, the authors have done considerable work both in vivo and in vitro. However, several issues are needed to be addressed before further consideration.

Thank you very much for your comments. We would like to answer your concerns point-by-point as below.

In light of the comments from reviewers, we have added more data and changed the structure of the revised manuscript without any changing in the flow of the story. Thus, all the figures quoted in our answers below are in the order of the revised manuscript.

1. The authors demonstrated that TfR1 expression was higher in the central MES region. However, in figure 1E, it seems that the expression of TfR1 in the margin part is higher than center part. Please confirm that the conclusion is consistent with the result.

Note that Fig. 1e has been changed to Fig. 2c of the revised manuscript.

We appreciate the need of clarification here. In Fig. 2a, we conducted an immunohistochemistry (IHC) experiment to check for the expression of proteins of interest in the margin (considered proneural) and center (considered mesenchymal) regions of glioblastoma patient's tissues.

In our IHC, we used 3,3'-Diaminobenzidine (DAB) as chromogen representing protein expression and Hematoxylin for nucleus staining. As shown in the figure, TF expression was higher in the margin with high brown signal (GBM1211 panel, first column) while TfR1 expression was higher in the center part with also high brown signal (GBM1211 panel, second column).

For this revision, we have attempted to get another set of samples with similar anatomical dissection and confirmed that the margin region expressed higher TfR1 (*Fig. 2b (GBM2021) with description added to lines 121-127 of the revised manuscript*).

2. The authors demonstrated that dopamine enhances ferroptosis of GBMs and sorafenib could mediate system Xc inhibition. And they proved that combined treatment with sorafenib and DA caused a further increase in overall survival compared with sorafenib alone. As system Xc plays important role in ferroptosis and sorafenib could induce ferroptosis by inhibiting it, please prove that system Xc was truly inhibited by sorafenib.

We appreciate your interesting question. We executed a cystine uptake assay upon sorafenib treatment in glioblastoma cells to confirm that sorafenib can truly inhibit system Xc by following a protocol as previously published ⁶. Briefly, cystine in the culture media was replaced by selenocystine which generates fluorescence signal with probes fluorescein O,O'-diacrylate (FOdA) and tris(2-carboxyethyl)phosphine (TCEP). The fluorescence signal intensity thus represents cystine uptake capacity. The result showed that sorafenib treatment did reduce cystine uptake in both two cell line X02 and 83 (*Rebuttal fig. 8, as Supplementary Fig. 6 with description added in lines 234-236 and relevant method added in lines 435-442 of the revised manuscript*).

Rebuttal figure 8. Cystine uptake assay upon sorafenib (1 μ M) treatment. Sorafenib reduced cystine uptake in GSCs.

3. GPX4 is a crucial protein in ferroptosis. Whether combine treatment with sorafenib and dopamine would influence its' expression level when compared with treatment with sorafenib or dopamine alone?

We thank for this comment. We examined the expression of GPX4 *in vivo* by using MES 83 tumor tissues from stereotactic mouse model. The results showed that GPX4 expression was reduced in the combined treatment of DA and sorafenib compared to each single treatment (Rebuttal fig. 9).

Rebuttal figure 9. Immunohistochemistry for expression of GPX4 under treatment of DA, sorafenib, or DA plus sorafenib. Combined DA and sorafenib reduced GPX4 protein.

4. Whether the susceptibility to ferroptosis of MES GSCs enhanced by dopamine could be reversed by ferroptosis inhibitor?

Thank you for this critical comment. To answer for the question, we treated MES GSCs with DA in combination with ferroptosis inducer RSL3 and tried to rescue with classical ferroptosis inhibitor Fer-1. The data showed that ferroptosis in MES GSCs enhanced by DA could be reversed by Fer-1 (Rebuttal fig. 10) indicating DA-promoted cell death here was in ferroptosis manner.

Rebuttal figure 10. MTS assay for cell viability under treatment of ferroptosis inhibitor together with RSL3 and DA. Fer-1 (2 μ M) showed rescue effect on the cell death induced by DA (20 μ M) in combination with RSL3 (1 μ M).

Reviewer #3

In this current manuscript, the authors are trying to address essential issues pretending to develop effective therapies against GBM, which is how the molecular heterogeneity promotes oncogenic progression. The authors here demonstrated that proneural GSCs could secrete dopamine and transferrin, which induced the proliferation of mesenchymal GSC and promoted gliomagenesis. This manuscript has several strengths, including investigating a critical issue in aggressive disease, using the patient-derived GSC model, and appropriate interpretation of the data. However, the manuscript also has some following significant weaknesses.

Thank you very much for your comments. We would like to answer your concerns point-by-point as below.

In light of the comments from reviewers, we have added more data and changed the structure of the revised manuscript without any changing in the flow of the story. Thus, all the figures quoted in our answers below are in the order of the revised manuscript.

1. As the author indicated, GBM is highly heterogeneous, with cells that can transit between different molecular states (PMID:24925914). Moreover, a recent report demonstrated that GBM cells could be in a hybrid state and acquire other states based on microenvironmental cues. One of the issues with this manuscript's experimental setup is the investigator utilizing bulk RNA-seq data that ignores the heterogeneity at the single-cell level while trying to study the intertumoral heterogeneity. It will be essential to investigate intertumoral heterogeneity because many different subtypes can be present in the same tumor. One such experiment should be carried out where authors co-culture genetically labeled (GFP or RFP) different subtypes of GSC and investigate how such interaction can influence the growth of different subtypes of GBM.

We appreciate your insightful idea with comment. Although we have intensively worked on, we unfortunately were not able to successfully establish GFP or RFP labeled GSCs.

Thus, to address your question, we have performed alternative experiments to directly co-culture MES and PN cells and check for growth rate. In one model, we utilized the permeable co-culture insert that allows cell-cell communication ([https://ecatalog.corning.com/life-sciences/b2c/US/en/Permeable-Supports/Inserts/Falcon%AE-Cell-Culture-Inserts/p/353493](https://ecatalog.corning.com/life-sciences/b2c/US/en/Permeable-Supports/Inserts/Falcon%C2%AE-Cell-Culture-Inserts/p/353493)) and culture one subtype in the bottom and the other subtype on the top. The data showed that MES cells increased growth capacity while PN cells were unaffected in co-culture condition (Rebuttal fig. 11).

In another model, we labelled each subtype with different cell tracing fluorescence probes (CMTPX red for X02 and Hoechst dye for 83) and co-cultured them in the same well. Consistently, MES cells grew faster in the presence of PN and PN had the tendency to surround MES cells (Rebuttal fig. 12). Taken together, we have validated our data showing commensalism symbiosis when PN supported MES growth using multiple approaches of co-culturing. These data will be added to *Fig. 1b with description in lines 79-83 and method in lines 332-333 of the revised manuscript.*

Rebuttal figure 11. Co-culture of PN and MES cells using insert. MES 83 increased cell viability while being co-cultured with PN X02.

Rebuttal figure 12. Co-culture of probe labeled-PN and MES. MES 83 grew faster in the presence of PN X02.

2. A global pathway enrichment analysis is necessary for genes enriched in different subtypes. These resources are open source and should be implemented to evaluate differentially regulated genes (Suppl. Fig. 1C and D) and their relation to other pathways.

We greatly appreciate your suggestion. We have performed a gene set enrichment analysis (GSEA) and a gene ontology (GO) analysis to evaluate the meaning of gene expression in different subtypes in IVY GAP database.

GSEA was carried out using IVY GAP datasets acquired from Affymetrix HT_HG-U133A microarray chip platform⁷ to distinguish the gene sets enriched in proneural regions (leading edge LE, and infiltrating tumor IT) from the gene sets in mesenchymal regions (pseudopalisading cells around necrosis CTpan, and microvascular proliferation CTmvp) of the tumor⁸. Using Hallmark and Gene Ontology Biological Process gene set databases from GSEA_4.0.3 (<https://www.gsea-msigdb.org/gsea/index.jsp>), we found that hypoxia, epithelial_mesenchymal_transition pathways were enriched in MES (Rebuttal fig. 13), whereas dopamine related pathways seemed to be upregulated in PN (Rebuttal fig. 14). These results were in accordance with our data showing the secretion of DA exclusively from PN to support iron uptake in MES and maintain MES growth under hypoxic condition. Enrichment plots of dopamine_secretion and dopamine_transport will be placed in *Supplementary Fig. 1e with description of results in lines 106-108 and method in lines 375-379 of the revised manuscript.*

Rebuttal figure 13. GSEA for IVY-GAP database. Hypoxia (left) and epithelial_mesenchymal_transition (right) pathways were enriched in MES regions.

Rebuttal figure 14. GSEA for IVY-GAP database. Synaptic_transmission_dopaminergic (left), dopamine_secretion (middle), and dopamine_transport pathways were enriched in PN regions.

Next, GO analysis was performed using web application GOnet (<https://tools.dice-database.org/GOnet/>)⁹.

We first determined differentially expressed genes between PN and MES regions with a significance of $p < 0.05$ from Supplementary fig. 1c and d and imported those genes into the GOnet server. The tables below show lists of the most significantly related pathways.

The results showed that genes including CA9, PDK1, SLC2A1, and VEGFA were highly expressed in MES, and related to the pathways regulating cellular response to oxygen levels or hypoxia (Rebuttal table 1), which confirmed again the hypoxic nature of MES glioblastoma.

Rebuttal table 1. GO analysis of genes associated with hypoxia in MES.

No.	GO_term_ID	GO_term_def	P_FDR_adj	Genes
1	GO:0001666	response to hypoxia	4.68E-04	CA9 PDK1 SLC2A1 VEGFA
2	GO:0036293	response to decreased oxygen levels	4.68E-04	CA9 PDK1 SLC2A1 VEGFA
3	GO:0070482	response to oxygen levels	4.68E-04	CA9 PDK1 SLC2A1 VEGFA
4	GO:0071456	cellular response to hypoxia	8.39E-03	CA9 PDK1 VEGFA
5	GO:0036294	cellular response to decreased oxygen levels	8.39E-03	CA9 PDK1 VEGFA
6	GO:0071453	cellular response to oxygen levels	8.98E-03	CA9 PDK1 VEGFA
7	GO:0009628	response to abiotic stimulus	1.74E-02	CA9 PDK1 SLC2A1 VEGFA

In addition, differentially expressed genes involved in iron metabolism (Supplementary fig. 1d, upper) were also associated to metal ion homeostasis or other specific pathways of iron transport or response to iron ion (Rebuttal table 2).

Rebuttal table 2. GO analysis of genes associated with iron metabolism.

No.	GO_term_ID	GO_term	P_FDR_adj	Genes
1	GO:0046916	cellular transition metal ion homeostasis	1.11E-08	FTL SLC11A2 SLC40A1 TF TFRC
2	GO:0000041	transition metal ion transport	1.74E-08	FTL SLC11A2 SLC40A1 TF TFRC
3	GO:0055072	iron ion homeostasis	3.50E-09	FTL SLC11A2 SLC40A1 TF TFRC
4	GO:0055076	transition metal ion homeostasis	2.09E-08	FTL SLC11A2 SLC40A1 TF TFRC
5	GO:0006879	cellular iron ion homeostasis	2.80E-09	FTL SLC11A2 SLC40A1 TF TFRC
6	GO:0006826	iron ion transport	2.80E-09	FTL SLC11A2 SLC40A1 TF TFRC
7	GO:0034755	iron ion transmembrane transport	5.55E-06	SLC11A2 SLC40A1 TF
8	GO:0006875	cellular metal ion homeostasis	2.23E-05	FTL SLC11A2 SLC40A1 TF TFRC
9	GO:0055065	metal ion homeostasis	3.09E-05	FTL SLC11A2 SLC40A1 TF TFRC
10	GO:0030003	cellular cation homeostasis	3.09E-05	FTL SLC11A2 SLC40A1 TF TFRC
11	GO:0030001	metal ion transport	3.09E-05	FTL SLC11A2 SLC40A1 TF TFRC

12	GO:0006873	cellular ion homeostasis	3.09E-05	FTL SLC11A2 SLC40A1 TF TFRC
13	GO:0010039	response to iron ion	3.58E-05	SLC11A2 SLC40A1 TF
14	GO:0055080	cation homeostasis	4.10E-05	FTL SLC11A2 SLC40A1 TF TFRC
15	GO:0098771	inorganic ion homeostasis	4.14E-05	FTL SLC11A2 SLC40A1 TF TFRC
16	GO:0055082	cellular chemical homeostasis	4.60E-05	FTL SLC11A2 SLC40A1 TF TFRC
17	GO:0050801	ion homeostasis	5.84E-05	FTL SLC11A2 SLC40A1 TF TFRC
18	GO:0006812	cation transport	7.60E-05	FTL SLC11A2 SLC40A1 TF TFRC
19	GO:0019725	cellular homeostasis	1.04E-04	FTL SLC11A2 SLC40A1 TF TFRC
20	GO:0048878	chemical homeostasis	2.78E-04	FTL SLC11A2 SLC40A1 TF TFRC
21	GO:0006811	ion transport	7.93E-04	FTL SLC11A2 SLC40A1 TF TFRC
22	GO:0060586	multicellular organismal iron ion homeostasis	9.22E-04	SLC11A2 SLC40A1
23	GO:0071281	cellular response to iron ion	1.10E-03	SLC40A1 TF
24	GO:0042592	homeostatic process	1.67E-03	FTL SLC11A2 SLC40A1 TF TFRC
25	GO:0033572	transferrin transport	1.42E-02	TF TFRC

GO analysis also showed that genes of dopamine receptors and synthesis enzymes (Supplementary fig. 1d, lower) were associated to catechol-containing compound metabolic process as well as signaling pathways (Rebuttal table 3).

Rebuttal table 3. GO analysis of genes associated with dopamine metabolism.

No.	GO_term_ID	GO_term	P_FDR_adj	Genes
1	GO:0009712	catechol-containing compound metabolic process	5.86E-07	DDC DRD1 DRD2 PAH
2	GO:0006584	catecholamine metabolic process	5.86E-07	DDC DRD1 DRD2 PAH
3	GO:0060158	phospholipase C-activating dopamine receptor signaling pathway	9.97E-07	DRD1 DRD2 DRD5
4	GO:0018958	phenol-containing compound metabolic process	3.96E-06	DDC DRD1 DRD2 PAH
5	GO:0043279	response to alkaloid	1.07E-05	DDC DRD1 DRD2 DRD5

6	GO:0001963	synaptic transmission, dopaminergic	1.38E-05	DRD1 DRD2 DRD5
7	GO:0042417	dopamine metabolic process	4.46E-05	DDC DRD1 DRD2
8	GO:0001975	response to amphetamine	5.29E-05	DRD1 DRD2 DRD5
9	GO:0007212	dopamine receptor signaling pathway	6.20E-05	DRD1 DRD2 DRD5
10	GO:0014075	response to amine	1.29E-04	DRD1 DRD2 DRD5
11	GO:0097366	response to bronchodilator	1.61E-04	DRD1 DRD2 DRD5
12	GO:0042220	response to cocaine	1.86E-04	DRD1 DRD2 DRD5
13	GO:0050805	negative regulation of synaptic transmission	2.48E-04	DRD1 DRD2 DRD5
14	GO:0001505	regulation of neurotransmitter levels	3.66E-04	DDC DRD1 DRD2 PAH
15	GO:0072347	response to anesthetic	3.98E-04	DRD1 DRD2 DRD5
16	GO:0008306	associative learning	3.98E-04	DRD1 DRD2 DRD5
17	GO:1903351	cellular response to dopamine	3.98E-04	DRD1 DRD2 DRD5
18	GO:0021853	cerebral cortex GABAergic interneuron migration	3.98E-04	DRD1 DRD2
19	GO:1904936	interneuron migration	3.98E-04	DRD1 DRD2
20	GO:0046958	nonassociative learning	3.98E-04	DRD1 DRD5
21	GO:1903350	response to dopamine	3.98E-04	DRD1 DRD2 DRD5
22	GO:0035690	cellular response to drug	3.98E-04	DDC DRD1 DRD2 DRD5
23	GO:1903831	signal transduction involved in cellular response to ammonium ion	4.64E-04	DRD1 DRD2 DRD5
24	GO:0007200	phospholipase C-activating G protein-coupled receptor signaling pathway	4.73E-04	DRD1 DRD2 DRD5
25	GO:0021894	cerebral cortex GABAergic interneuron development	4.73E-04	DRD1 DRD2

Overall, GO analysis data provide the pathways associated to iron and dopamine related genes shown in our manuscript.

3. The discrepancy in DRD1 mRNA expression clinically (Suppl Fig 1D) and experimentally (Fig 1D) is troubling. Authors should consider using other public sources of GBM patient data, such as <https://www.proteinatlas.org> and <http://gliovis.bioinfo.cnio.es>, to validate their findings. Moreover, it would be critical to investigate all the DRD receptors in patient's GBM in Fig 1e.

We appreciate your concern. We have searched for data from The Human Protein Atlas (<https://www.proteinatlas.org>) and Gliovis (<http://gliovis.bioinfo.cnio.es>) projects. While we were not able to formulate protein expression of DRD receptors in glioblastoma tissues from The Human Protein Atlas, various datasets allowed us to distinguish gene expression between PN and MES subtypes of glioblastoma from the Gliovis. Not only DRD1, but we also checked for other relevant genes of interest. In general, supportive to our data in the manuscript, PN expressed higher transferrin gene (TF), dopamine receptors (DRD5, DRD1), and dopamine synthesis enzymes (TH, PAH, DDC) whereas MES upregulated transferrin receptor gene (TFRC) (Rebuttal fig. 15 – 21, as *Supplementary Fig. 2b, c with result sentences in lines 131-134 and method in lines 373-374 of the revised manuscript*).

Rebuttal figure 15. Expression of **TF** in multiple glioblastoma datasets. TF was upregulated in PN compared to MES.

Rebuttal figure 16. Expression of **TFRC** in multiple glioblastoma datasets. TFRC was upregulated in MES compared to PN.

Rebuttal figure 17. Expression of **DRD5** in multiple glioblastoma datasets. DRD5 was upregulated in PN compared to MES.

Rebuttal figure 18. Expression of **DRD1** in multiple glioblastoma datasets. DRD1 was upregulated in PN compared to MES.

Rebuttal figure 19. Expression of **TH** in multiple glioblastoma datasets. TH was upregulated in PN compared to MES.

Rebuttal figure 20. Expression of **PAH** in multiple glioblastoma datasets. PAH was upregulated in PN compared to MES.

Rebuttal figure 21. Expression of **DDC** in multiple glioblastoma datasets. PAH was upregulated in PN compared to MES.

Secondly, as you suggested, we have investigated **all the DRD receptors** in patient's glioblastoma tissue. The immunohistochemistry data showed low or no expression of DRD1, DRD2, DRD3, and DRD4 in both two regions margin and center of the tumor (Rebuttal fig. 22, as *Supplementary Fig. 2a with result in lines 127-128 of the revised manuscript*). This data suggested a specific and important function of DRD5 in glioblastoma pathology.

Rebuttal figure 22. Expression of DRD1, DRD2, DRD3, and DRD4 in margin and center regions of patient's glioblastoma tissue. IHC data showed no expression of these DRD receptors.

4. Validating the finding in Fig 1E in multiple GBM tissues would be critical.

Please be noted again that Fig. 1e has been changed to Fig. 2b of the revised manuscript.

Thank you for your suggestion. Accordingly, we have tried to obtain more glioblastoma tissues that anatomically dissected to margin and center regions similar to what we showed in the current manuscript. The number of samples is tightly dependent on the availability of the patients, surgery, and surgeons. Over the time of this revision, we were able to get only one more set of samples.

We conducted the same experiments to check for the expression of TF, TfR1, DRD5, TH, PAH, and DDC, and as you suggested in question 5, we also extended to all other DRD receptors including DRD1, DRD2, DRD3, DRD4. Our data showed consistently that transferrin receptor TfR1 were highly expressed in the center region while transferrin, dopamine receptors DRD5, and dopamine synthesis enzymes seemed to express in margin region (Rebuttal fig. 23, as *Fig. 2b, Supplementary Fig. 2a with result description in lines 121-128 of the revised manuscript*).

Rebuttal figure 23. Expression of proteins of interest in margin and center regions of patient's glioblastoma tissue. IHC data showed the strong expression of Tfr1 in center region; TF, DRD5 and DA synthesis enzymes in the margin region.

5. Fig. 1F, the IF picture should be captured in higher magnification. It would be essential to capture images in multiple areas digitally analyze by ImageJ each picture for expression. From the current images presented in the manuscript, it is unclear how these images represent the total disease area.

Please be noted again that Fig. 1f has been changed to Fig. 2c of the revised manuscript.

We thank you for this suggestion. Accordingly, we captured multiple images in higher magnification from PN and MES regions, and digitally analyzed for the fluorescence density which represents protein expression using ImageJ. The quantification data will be put in parallel with the images to make clearer the association of PN regions with DRD5 and TF as well as MES regions with Tfr1 expression (Rebuttal fig. 24, as Fig. 2d with method described in lines 401-402 of the revised manuscript).

Rebuttal figure 24. Quantification of protein expression in human glioblastoma tissues shown in Fig. 2b.

6. Line 126-127, MES originated in hypoxia, and please add a reference that supports this statement.

We appreciate your comment. We added the reference supporting that hypoxic region of glioblastoma harbor mesenchymal characteristics (Nat Med. 2017 Nov; 23(11): 1352–1361)⁸ as well as hypoxic condition promoted MES initiation (Cancer Lett. 2015 Apr 1;359(1):107-16 and Cell Death Dis. 2020 Mar 3;11(3):168)^{10,11}. *These references were added in line 144 of the revised manuscript.*

7. It is unclear if the TF and Fe concentration used in the experimental setting is physiologically relevant. Why did the authors choose to use this concentration? Would you please justify using references?

We appreciate your comments. To rationally clarify this concern, we have checked multiple references for proper concentration of Fe and TF used in *in vitro* experimental setting, especially in the field of cancer research.

TF concentration

TF has been used with concentration varying among studies, from 0.1 $\mu\text{g/ml}$ ¹² to 200 $\mu\text{g/ml}$ ¹³. The high dose as 100 $\mu\text{g/ml}$ was proved to be nontoxic¹⁴. Typical doses were 7 $\mu\text{g/ml}$ ¹⁵, 10 $\mu\text{g/ml}$ ¹², 20 $\mu\text{g/ml}$ ¹⁵, 25 $\mu\text{g/ml}$ ¹⁶, 50 $\mu\text{g/ml}$ ^{13,17}. Thus, the dose of TF used in our study (20 $\mu\text{g/ml}$) were within the range of doses justified in references.

Fe concentration

It has been agreed that glioblastoma has high demand on iron, however, information on serum level of iron in glioblastoma patients is still not conclusive¹⁸. Meanwhile, the physiological concentration of iron in the plasma is in the range of **10-30 μM** ¹⁹, or approximately **18.4 \pm 5.9 μM** ²⁰, which could be increased to approximately **54 μM** in cancer patients²¹. Concentration of iron (in form of ferrous or ferric) used to treat in cell culture has been varied from as low as 2 μM ²² to 5-6 mM^{23,24}. Thus, in most experiments we

have treated 18 μ M which is physiologically relevant and in range of doses used in other references. The doses were incremented or decremented around the base of 18 μ M in some experiments checking dose-dependent phenomenon.

8. According to Fig 2f, TF and Fe can stabilize Hif selectively in the MES lines, and authors claim that such stability may be responsible for increased growth observed in Fig 2a. However, it is not explained by the authors why we also kept increased growth in the normoxic conditions in Fig 2A as both TF and Fe didn't alter the HIF expression in Fig 2a. is it possible that increased growth HIF independent?

Please be noted again that Fig. 2 has been changed to Fig. 3 of the revised manuscript and the figures quoted below are in the order of the revised manuscript.

We greatly appreciate your valuable comment. We would discuss as follows: Both TF and Fe activate Src-ERK signaling under normoxia and hypoxia in MES cells. Thus, activation of Src-ERK lead to the increased growth as it is (Fig. 3a, 83, second bar compared to first) while activation of Src-ERK further stabilizes Hif1 α (which was only activated in hypoxia but not normoxia) to rescue cell growth (Fig. 3a, 83, the forth bar compared to third) that is typically decreased under hypoxia (Fig. 3a, 83, third bar compared to first). Additional experiments and explanation were below.

Iron signaling had potentially reciprocal links to Src pathway^{25,26}. We also showed that, the downstream factor of iron signaling in MES would be oncogenic Src (Fig. 3c, f) and its downstream ERK but not STAT3 (Supplementary fig. 3b).

Since MES is associated with hypoxia^{8,10,11}, and Src-ERK signaling was important for maintaining hypoxia-inducible factor Hif1 α ²⁷⁻²⁹, we questioned the function of iron-Src-ERK on MES growth under such condition. Consistently, we observed iron dependent Hif1 α stability through Src activation upon TF or iron treatment in MES (Fig. 3f).

To deepen and clarify this mechanism, in this revision, we treated MES GSCs with two different types of Src inhibitors SU6656 and PP2. Interestingly, PP2 suppressed pERK while SU6656 only inhibited pSTAT3 (Rebuttal fig. 25). Consistently, PP2 but not SU6656 was able to abolish Hif1 α expression stabilized by TF or iron treatment under hypoxia (Rebuttal fig. 26, as *Supplementary Fig. 3d with description of the result in lines 170-172 of the revised manuscript*), supporting the specificity of Src-ERK-Hif1 α axis activated by iron. The improvement of MES proliferation by TF or iron treatment in both normoxia and hypoxia was abolished by PP2 as well (Rebuttal fig. 27, as *Fig. 3g with description of result in lines 172-173 of the revised manuscript*).

Rebuttal figure 25. Inhibition of Src by SU6656 or PP2. Treatment of SU6656 (5 μ M) or PP2 (20 μ M) differently suppressed Src downstreams pSTAT3 and pERK, respectively.

Rebuttal figure 26. Regulation of iron-Src-Erk signaling pathway by Src inhibitor. PP2 (right), but not SU6656 (left) inhibited pERK thus reduced Hif1 α stabilized by TF or iron treatment.

Rebuttal figure 27. MES GSCs cell growth upon Src inhibitor treatment. PP2 abolished TF- or iron-induced cell viability both in normoxia and hypoxia.

9. As the authors mentioned, there are conflicting reports published in the literature regarding the specific DRD receptors' involvement in gliomagenesis. Some reports indicated that DRD3-4 is critical (PMID:32358191). Others indicated that maybe DRD2 is critical (PMID:30651332). Based on these conflicting results and the lack of specificity for some of the DRD inhibitors, it would be essential to demonstrate the importance of DRD5 by genetically knockdown the receptor and perform the same experiments in fig 3.

Please be noted again that Fig. 3 has been changed to Fig. 4 of the revised manuscript and the figures noted below are in the order of the revised manuscript.

We appreciate your valuable comments. In Fig. 4, we have shown that DA treatment increases pSrc exclusively in PN. This effect is possibly through DRD5 as the pharmaceutical inhibitor of the receptor inhibits pSrc (Supplementary fig. 4a) and thus suppresses PN growth (Fig. 4b). As the reviewer's suggestion, we executed the knockdown experiment in PN and MES GSCs using siRNA against DRD5. The data show that DRD5 knockdown reduces PN 528 cells growth (Rebuttal fig. 28), by abolishing DA-induced pSrc (Rebuttal fig. 29). Thus, DRD5 is the critical receptor for the DA effect in PN GSC. *These data were added to Supplementary Fig. 4b, c with description in lines 193-195 and method in lines 338-342 of the revised manuscript.*

Rebuttal figure 28. Knockdown of DRD5 in GSCs. MTS assay showed reduction of PN 528 cell viability upon DRD5 knockdown.

Rebuttal figure 29. Knockdown of DRD5 in GSCs. Western blot showed that DRD5 knockdown abolished DA effect on pSrc in PN 528.

10. Please provide a scientific rationale for the use of ferroptosis induced in fig 4. Is this therapeutic approach will work in combination with other therapies?

Note that Fig. 4 has been changed to Fig. 5 of the revised manuscript.

We are not sure if we could understand this question correctly. We here try to answer for the rationale ferroptosis inducer used in the figure you mentioned, and clinical implication with other therapies.

We made a strategy of inducing ferroptosis in GSCs by different approaches, including cystine deprivation, treatment of erastin, RSL3, and sorafenib³⁰, in single treatment or in combination with DA. Such strategy firstly aimed to prove and validate that our hypothesis was applicable for various ferroptosis inducers.

Ferroptosis is an iron-dependent, lipid peroxidation-induced type of cell death which can be triggered by the disruption of the cellular lipid peroxidation scavenging system consisting of cystine - cystine transporter Xc – glutathione - glutathione peroxidase 4 (GPX4) axis³¹.

Cystine deprivation is the direct strategy to starve the cancer cells from cystine and has been reported earlier for cancer growth inhibitory effect both *in vitro* and *in vivo*^{32,33}. Cystine deprivation decreased glutathione synthesis and induced gliomas cell death and prolonged gliomas-bearing mice survival³⁴.

Erastin, and RSL3 both have been used widely to induce ferroptosis since they were proposed as system Xc inhibitor and GPX4 inhibitor, respectively^{2,31,35}.

Sorafenib is known as a multi-kinase inhibitor which recently further identified to inhibit system Xc function^{36,37}. We also confirmed that sorafenib could decrease cellular cystine using a fluorescence-based cystine uptake assay (*Supplementary Fig. 6 with description added in lines 234-236 and relevant method added in lines 435-442 of the revised manuscript*). Based on the literature evidence and our data, sorafenib use was rationalized for the stereotactic brain tumor model given its ability to cross the blood-brain barrier³⁸.

These ferroptosis inducers have been well studied in combination with other therapies to overcome drug resistance. For instance, combination with cetuximab or genetic silence of prominin2 (PROM2) could sensitize cancer cells to RSL3^{39,40}. In another report, AZD-8055 or Nedd4 knockout both promoted erastin-induced ferroptosis^{4,5}. Sorafenib could be used together with artesunate or ABCC5 knockout in hepatocellular carcinoma for stronger ferroptosis ignition^{41,42}.

Specifically, in glioblastoma, it has been suggested that ferroptosis inducers could work with other therapies. Erastin has been shown to work in cooperation with temozolomide, a classical chemotherapy of glioblastoma^{43,44}. Another study proposed that RSL3 should be combined with NF-κB inhibitor BAY 11-7082 for an optimal induction of glioblastoma cell death in ferroptotic manner⁴⁵. In this manuscript, we

proposed another promising combination of DA with ferroptosis inducers that synergistically enhanced ferroptosis in glioblastoma, specifically aggressive mesenchymal subtype.

11. Fig 4a, the therapeutic effect is not selective to MES; the viability of the proneural GSC is significantly reduced in the Veh condition. Why?

Please be noted again that Fig. 4a has been changed to Fig. 5a.

We appreciate your point. We apologize not clear indication of “the Veh” in the figure for the reviewer’s question. The “Veh” actually means ‘Cystine deprivation’ and fixed to be clearer. Again, it has been described in the text that “After observing increased iron uptake into MES GSCs mediated via the paracrine effects of PN GSC-secreted DA and TF, we sought to investigate the biological consequence of intracellular iron accumulation and further link to potential therapeutic strategies for MES GBM. To this end, the GSC panel was pharmacologically assessed for ferroptosis induction. **Interestingly, PN GSCs, but not MES GSCs, became susceptible to various ferroptotic stresses**, such as RSL3-mediated GPX4 inhibition, erastin- and sorafenib-mediated system Xc inhibition, and culture medium cystine deprivation, and the effects of which were reversed via ferrostatin-1-induced inhibition of lipid peroxidation. **Interestingly, MES GSCs became susceptible to ferroptotic stress upon DA treatment**, which might be associated with increased iron uptake in MES GSCs and the accumulated labile iron pool in both subtypes, as described earlier”.

12. The Subcutaneous tumor model in Fig 4e does not provide any added information for developing therapy in GBM. I would recommend removing this fig in the supplementary fig.

We thank you for this comment and move the data from Fig. 4e to the *Supplementary Fig. 7 with description in lines 238-244 of the revised manuscript.*

13. Fig 4f should be separated by PN and MES subtypes as this manuscript's main hypothesis address the molecular heterogeneity of the GBM.

Please be noted again that Fig. 4f has been changed to Fig. 5e of the revised manuscript.

We apologize for confusing this data. As we initially showed that proneural subtype was characterized by high expression of TF and DRD5 while the mesenchymal subtype was high in TFRC (this phenomenon was also observed in TCGA dataset, please refer to the Rebuttal fig. 13 and 14), we questioned if these expressions would represent the survival of patients who had mixed subtypes (which means HIGH expression of both TF-TFRC or DRD5-TFRC). Thus, we classified all the patients in the TCGA dataset (regardless of their dominant subtypes PN or MES) based on median expression of TF, DRD5, and TFRC. Indeed, we found that patients having high expression of both genes associated with proneural and mesenchymal may have the shortest survival time. This survival data was in agreement with our molecular data showing functions of TF, TFRC, and DRD5 in heterogenous symbiosis of proneural and mesenchymal cancer cells which elevated cancer growth.

We noticed that our text description would not be enough to clarify the purpose of analysis, as your comment suggested. We would like to add more explanation in *lines 244-255 of the revised manuscript*.

References

- 1 Conrad, M. *et al.* Regulation of lipid peroxidation and ferroptosis in diverse species. *Genes Dev* **32**, 602-619, doi:10.1101/gad.314674.118 (2018).
- 2 Dixon, S. J. *et al.* Ferroptosis: an iron-dependent form of nonapoptotic cell death. *Cell* **149**, 1060-1072, doi:10.1016/j.cell.2012.03.042 (2012).
- 3 Juan, C. A., Perez de la Lastra, J. M., Plou, F. J. & Perez-Lebena, E. The Chemistry of Reactive Oxygen Species (ROS) Revisited: Outlining Their Role in Biological Macromolecules (DNA, Lipids and Proteins) and Induced Pathologies. *Int J Mol Sci* **22**, doi:10.3390/ijms22094642 (2021).
- 4 Yang, Y. *et al.* Nedd4 ubiquitylates VDAC2/3 to suppress erastin-induced ferroptosis in melanoma. *Nat Commun* **11**, 433, doi:10.1038/s41467-020-14324-x (2020).

- 5 Zhang, Y. *et al.* mTORC1 couples cyst(e)ine availability with GPX4 protein synthesis and ferroptosis regulation. *Nat Commun* **12**, 1589, doi:10.1038/s41467-021-21841-w (2021).
- 6 Shimomura, T. *et al.* Simple Fluorescence Assay for Cystine Uptake via the xCT in Cells Using Selenocystine and a Fluorescent Probe. *ACS Sens* **6**, 2125-2128, doi:10.1021/acssensors.1c00496 (2021).
- 7 Bronisz, A., Salinska, E., Chiocca, E. A. & Godlewski, J. Hypoxic Roadmap of Glioblastoma-Learning about Directions and Distances in the Brain Tumor Environment. *Cancers (Basel)* **12**, doi:10.3390/cancers12051213 (2020).
- 8 Jin, X. *et al.* Targeting glioma stem cells through combined BMI1 and EZH2 inhibition. *Nat Med* **23**, 1352-1361, doi:10.1038/nm.4415 (2017).
- 9 Pomaznoy, M., Ha, B. & Peters, B. GOnet: a tool for interactive Gene Ontology analysis. *BMC Bioinformatics* **19**, 470, doi:10.1186/s12859-018-2533-3 (2018).
- 10 Wang, S. *et al.* Hypoxia-induced lncRNA PDIA3P1 promotes mesenchymal transition via sponging of miR-124-3p in glioma. *Cell Death Dis* **11**, 168, doi:10.1038/s41419-020-2345-z (2020).
- 11 Joseph, J. V. *et al.* Hypoxia enhances migration and invasion in glioblastoma by promoting a mesenchymal shift mediated by the HIF1alpha-ZEB1 axis. *Cancer Lett* **359**, 107-116, doi:10.1016/j.canlet.2015.01.010 (2015).
- 12 Carlevaro, M. F. *et al.* Transferrin promotes endothelial cell migration and invasion: implication in cartilage neovascularization. *J Cell Biol* **136**, 1375-1384, doi:10.1083/jcb.136.6.1375 (1997).
- 13 Ono-Uruga, Y. *et al.* Human adipose tissue-derived stromal cells can differentiate into megakaryocytes and platelets by secreting endogenous thrombopoietin. *J Thromb Haemost* **14**, 1285-1297, doi:10.1111/jth.13313 (2016).

- 14 Chen-Roetling, J., Chen, L. & Regan, R. F. Apotransferrin protects cortical neurons from hemoglobin toxicity. *Neuropharmacology* **60**, 423-431, doi:10.1016/j.neuropharm.2010.10.015 (2011).
- 15 Fassl, S. *et al.* Transferrin ensures survival of ovarian carcinoma cells when apoptosis is induced by TNFalpha, FasL, TRAIL, or Myc. *Oncogene* **22**, 8343-8355, doi:10.1038/sj.onc.1207047 (2003).
- 16 Liang, W., Li, Q. & Ferrara, N. Metastatic growth instructed by neutrophil-derived transferrin. *Proc Natl Acad Sci U S A* **115**, 11060-11065, doi:10.1073/pnas.1811717115 (2018).
- 17 Tan, S. S. *et al.* Therapeutic MSC exosomes are derived from lipid raft microdomains in the plasma membrane. *J Extracell Vesicles* **2**, doi:10.3402/jev.v2i0.22614 (2013).
- 18 Legendre, C. & Garcion, E. Iron metabolism: a double-edged sword in the resistance of glioblastoma to therapies. *Trends Endocrinol Metab* **26**, 322-331, doi:10.1016/j.tem.2015.03.008 (2015).
- 19 Ganz, T. & Nemeth, E. Heparin and disorders of iron metabolism. *Annu Rev Med* **62**, 347-360, doi:10.1146/annurev-med-050109-142444 (2011).
- 20 Mojic, M. *et al.* Extracellular iron diminishes anticancer effects of vitamin C: an in vitro study. *Sci Rep* **4**, 5955, doi:10.1038/srep05955 (2014).
- 21 Bae, Y. J., Yeon, J. Y., Sung, C. J., Kim, H. S. & Sung, M. K. Dietary intake and serum levels of iron in relation to oxidative stress in breast cancer patients. *J Clin Biochem Nutr* **45**, 355-360, doi:10.3164/jcbtn.09-46 (2009).
- 22 Li, Z., Tanaka, H., Galiano, F. & Glass, J. Anticancer activity of the iron facilitator LS081. *J Exp Clin Cancer Res* **30**, 34, doi:10.1186/1756-9966-30-34 (2011).
- 23 Park, K. J. *et al.* Quantitative characterization of the regulation of iron metabolism in glioblastoma stem-like cells using magnetophoresis. *Biotechnol Bioeng* **116**, 1644-1655, doi:10.1002/bit.26973 (2019).

- 24 He, W. L., Feng, Y., Li, X. L., Wei, Y. Y. & Yang, X. E. Availability and toxicity of Fe(II) and Fe(III) in Caco-2 cells. *J Zhejiang Univ Sci B* **9**, 707-712, doi:10.1631/jzus.B0820023 (2008).
- 25 Cao, H., Chen, J., Krueger, E. W. & McNiven, M. A. SRC-mediated phosphorylation of dynamin and cortactin regulates the "constitutive" endocytosis of transferrin. *Mol Cell Biol* **30**, 781-792, doi:10.1128/MCB.00330-09 (2010).
- 26 Jian, J., Yang, Q. & Huang, X. Src regulates Tyr(20) phosphorylation of transferrin receptor-1 and potentiates breast cancer cell survival. *J Biol Chem* **286**, 35708-35715, doi:10.1074/jbc.M111.271585 (2011).
- 27 Lamberti, M. J. *et al.* Transcriptional activation of HIF-1 by a ROS-ERK axis underlies the resistance to photodynamic therapy. *PLoS One* **12**, e0177801, doi:10.1371/journal.pone.0177801 (2017).
- 28 Lee, H. Y. *et al.* Src activates HIF-1alpha not through direct phosphorylation of HIF-1alpha specific prolyl-4 hydroxylase 2 but through activation of the NADPH oxidase/Rac pathway. *Carcinogenesis* **32**, 703-712, doi:10.1093/carcin/bgr034 (2011).
- 29 Jung, F. *et al.* Hypoxic induction of the hypoxia-inducible factor is mediated via the adaptor protein Shc in endothelial cells. *Circ Res* **91**, 38-45, doi:10.1161/01.res.0000024412.24491.ca (2002).
- 30 Hassannia, B., Vandenabeele, P. & Vanden Berghe, T. Targeting Ferroptosis to Iron Out Cancer. *Cancer Cell* **35**, 830-849, doi:10.1016/j.ccell.2019.04.002 (2019).
- 31 Jiang, X., Stockwell, B. R. & Conrad, M. Ferroptosis: mechanisms, biology and role in disease. *Nat Rev Mol Cell Biol* **22**, 266-282, doi:10.1038/s41580-020-00324-8 (2021).
- 32 Badgley, M. A. *et al.* Cysteine depletion induces pancreatic tumor ferroptosis in mice. *Science* **368**, 85-89, doi:10.1126/science.aaw9872 (2020).

- 33 Cramer, S. L. *et al.* Systemic depletion of L-cyst(e)ine with cyst(e)inase increases reactive oxygen species and suppresses tumor growth. *Nat Med* **23**, 120-127, doi:10.1038/nm.4232 (2017).
- 34 Ruiz-Rodado, V. *et al.* Cysteine is a limiting factor for glioma proliferation and survival. *Mol Oncol*, doi:10.1002/1878-0261.13148 (2021).
- 35 Yang, W. S. *et al.* Regulation of ferroptotic cancer cell death by GPX4. *Cell* **156**, 317-331, doi:10.1016/j.cell.2013.12.010 (2014).
- 36 Lachaier, E. *et al.* Sorafenib induces ferroptosis in human cancer cell lines originating from different solid tumors. *Anticancer Res* **34**, 6417-6422 (2014).
- 37 Dixon, S. J. *et al.* Pharmacological inhibition of cystine-glutamate exchange induces endoplasmic reticulum stress and ferroptosis. *Elife* **3**, e02523, doi:10.7554/eLife.02523 (2014).
- 38 Siegelin, M. D., Raskett, C. M., Gilbert, C. A., Ross, A. H. & Altieri, D. C. Sorafenib exerts anti-glioma activity in vitro and in vivo. *Neurosci Lett* **478**, 165-170, doi:10.1016/j.neulet.2010.05.009 (2010).
- 39 Yang, J. *et al.* Cetuximab promotes RSL3-induced ferroptosis by suppressing the Nrf2/HO-1 signalling pathway in KRAS mutant colorectal cancer. *Cell Death Dis* **12**, 1079, doi:10.1038/s41419-021-04367-3 (2021).
- 40 Brown, C. W., Chhoy, P., Mukhopadhyay, D., Karner, E. R. & Mercurio, A. M. Targeting prominin2 transcription to overcome ferroptosis resistance in cancer. *EMBO Mol Med* **13**, e13792, doi:10.15252/emmm.202013792 (2021).
- 41 Li, Z. J. *et al.* Artesunate synergizes with sorafenib to induce ferroptosis in hepatocellular carcinoma. *Acta Pharmacol Sin* **42**, 301-310, doi:10.1038/s41401-020-0478-3 (2021).
- 42 Huang, W. *et al.* ABCC5 facilitates the acquired resistance of sorafenib through the inhibition of SLC7A11-induced ferroptosis in hepatocellular carcinoma. *Neoplasia* **23**, 1227-1239, doi:10.1016/j.neo.2021.11.002 (2021).

- 43 Chen, L. *et al.* Erastin sensitizes glioblastoma cells to temozolomide by restraining xCT and cystathionine-gamma-lyase function. *Oncol Rep* **33**, 1465-1474, doi:10.3892/or.2015.3712 (2015).
- 44 Sehm, T. *et al.* Temozolomide toxicity operates in a xCT/SLC7a11 dependent manner and is fostered by ferroptosis. *Oncotarget* **7**, 74630-74647, doi:10.18632/oncotarget.11858 (2016).
- 45 Li, S. *et al.* RSL3 Drives Ferroptosis through NF-kappaB Pathway Activation and GPX4 Depletion in Glioblastoma. *Oxid Med Cell Longev* **2021**, 2915019, doi:10.1155/2021/2915019 (2021).